# Latent Variable Identifiability in Nonlinear Causal Models with Single-domain Data under Minimality Condition

## Abstract

The identifiability of latent variables given observational data is one of the core issues in the field of disentangled representation learning. Recent progresses have been made on establishing identifiablity theories for latent causal models. However with much restrictions or unrealistic assumptions, their practicality on real applications are limited. In this paper, we propose a novel identifiablity theory for learning latent variables in nonlinear causal models, requiring only single-domain data. We prove that all latent variables in a powerset bipartite graph can be identified up to an invertible transformation, if the generation process of observable data is globally invertible, latent variables are independent, and shared latent variables entail minimal information. Experiments on synthetic data support the conclusions of our theory.

## 1 Introduction

Disentangled representation learning is one of the fundamental problems in machine learning (Bengio et al., 2013). It aims at discovering the underlying latent variables given a set of observed variables, which is essential in many downstream tasks such as domain adaptation (Baktashmotlagh et al., 2018; Cai et al., 2019a), out-of-distribution generalization (Sun et al., 2021; Chen et al., 2023), and style transfer (Gonzalez-Garcia et al., 2018; Lee et al., 2018). The unsupervised learning of disentangled representations, though common in practice, is however fundamentally impossible without inductive biases (Locatello et al., 2019). Therefore, whether latent variables can be uniquely determined (up to an equivalent class) under certain conditions, becomes a core issue in this field, which is usually referred to as *the identifiability problem of latent variables* (Khemakhem et al., 2020).

Early works on identifiability of latent variables mainly originate from the nonlinear ICA theory (Hyvärinen & Pajunen, 1999), which guarantees the existence of latent variable $\mathbf{s}$ with independent components $\mathbf{s} = \{s_i\}_{i=1}^n$ under nonlinear invertible transformation $\mathbf{s} = f(\mathbf{x})$ over observed variable $\mathbf{x}$. To ensure identifiability, further restrictions such as extra auxiliary variables (Hyvarinen & Morioka, 2016; Khemakhem et al., 2020) or fixed function class (Buchholz et al., 2022) are required. Traditional theories on ICA regard each dimension of $\mathbf{s}$ as an individual variable, however, latent factors in disentangled representation learning tasks are usually multi-dimensional rather than one-dimensional, leading to the requirement for block-wise identifiability.

Some recent works have tried to establish block-wise identifiability theory based on structural causal models (Pearl, 2009), aiming to identify the underlying generation process of observed data (Sun et al., 2021; Chen et al., 2023; Buchholz et al., 2024; Zhang et al., 2024). Such works still require extra auxiliary information, which relies on available multi-domain data. However, multi-domain data are usually hard to acquire (Matsuura & Harada, 2020; Creager et al., 2021), limiting the applicability of the theory. Other block-wise identifiability works make restrictive assumptions such as additive function (Lachapelle et al., 2024), compositionality (Brady et al., 2023; Wiedemer et al., 2023), or subspace span(Kong et al., 2024), which may only be suitable for specific scenarios.

In this paper, we aim to establish a new identifiability theory for nonlinear causal models with mild assumptions, while requiring only single-domain data. We first show that given a set of observed variables, any underlying structural causal model (SCM) can be reduced to an equivalent SCM whose structure is a powerset bipartite graph (PBG), while each latent variable in a PBG-SCM is equivalent

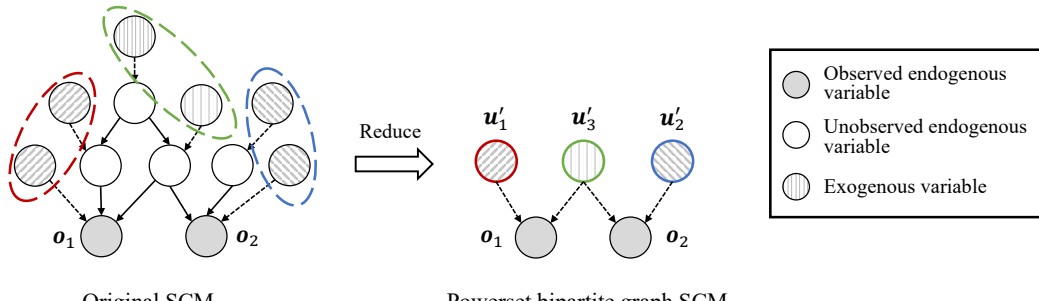

Figure 1: The proposed reduction process of a structural causal model. Our identifiability theory applies to latent causal models with powerset bipartite graph structures, which provides an equivalent class of general causal models under the reduction transformation.

to the concatenation of a set of exogenous variables in original SCM. We then prove that, given single-domain observation on observed variables, each latent variable in a PBG-SCM can be identified up to an invertible transformation under mild conditions. Our theory include 3 main assumptions: *invertibility* of global mapping, which guarantees the latent variables can be recovered from observed variables; *independence* of latent variables, which guarantees each latent variable contains unique information; *minimality* of shared latent variables, which guarantees individual information not be shared. While the first two assumptions are common in existing literatures, the minimality assumption is first proposed in this paper and plays a key role. All these assumptions are mild, and necessary for designing identification algorithms. Our main contributions include:

- We proposed a new identifiability theory with a novel minimality condition for latent variables in nonlinear causal models with single-domain data.

- We proposed a reduction process of structural causal models, establishing the connection between general causal graphs and bipartite graphs in terms of latent variable identification.

- We demonstrated through experiments that, to ensure successful identification of latent variables, it is necessary to include mechanisms corresponding to each assumptions in the algorithm design.

## 2 RELATED WORKS

**Disentangled representation learning.** Works on disentangled representation learning have discussed the property of ideal disentanglement, such as modularity, compactness, and explicitness (Ridgeway & Mozer, 2018; Higgins et al., 2018; Eastwood & Williams, 2018). While closely related to our identification goals and some of our assumptions, existing identifiability theory based on such objectives (Eastwood et al., 2022) is incapable to provide guidance for learning. Beside, there are lots of works focusing on practical algorithm to identify latent variables (Higgins et al., 2017; Chen et al., 2018; Ganin et al., 2016), but few of them have discussed the identifiability result, or include sufficient mechanism to guarantee the uniqueness of solution. Our work exactly provides a guidance for designing identification algorithms with theoretical support.

**Latent variable identifiability.** To guarantee the uniqueness of identified latent variables, there are two main branches of works for building the identifiability theory. One branch of existing works opts to introduce extra information (Khemakhem et al., 2020; Yang et al., 2022), such as data with domain index (Sorrenson et al., 2020; Sun et al., 2021; Kong et al., 2023), similar data from multiple views (Hyvarinen et al., 2019; Von Kügelgen et al., 2021; Gresele et al., 2020; Zimmermann et al., 2021), interventional data (Buchholz et al., 2024; Zhang et al., 2024), or data with time index (Hyvarinen & Morioka, 2016; Klindt et al., 2020; Lachapelle et al., 2022; Li et al., 2024). Acquiring such extra information may be difficult or impossible in many scenarios, so that another branch opts to fully utilize the existing single-domain data, at the cost of introducing other restrictions. Lachapelle et al. (2024) assume the transformation is additive across latent variables, which may be unrealistic for general learning tasks. Brady et al. (2023) and Wiedemer et al. (2023) make compositionality

assumptions for object-centric learning tasks, which is not applicable for latent variables with global influence. Kong et al. (2024) require the ground truth model and the learned model to jointly satisfy a subspace-span condition, which cannot judge whether it is satisfied by a single model. Our work also falls into the latter branch, while we propose a minimality assumption which is much easier to be satisfied in general scenarios.

**Causal structure identifiability.** Structure identification in latent causal models is a problem closely related to latent variable identification (Cai et al., 2019b; Kivva et al., 2021; Chen et al., 2022; Reizinger et al., 2023; Jiang & Aragam, 2024), and some works discuss these two problems together (Kong et al., 2024; Zhang et al., 2024). Our work do not aim to identify the structure of underlying SCM, since this will inevitably require additional assumptions. However, we do discuss the equivalent classes of graph structures, in a sense that if a latent variable with constant value is identified, then the corresponding vertex in the graph can be removed.

**Causal abstraction.** Causal abstraction literatures (Rubenstein et al., 2017; Beckers et al., 2020; Geiger et al., 2021; Xia & Bareinboim, 2024) study under what conditions that two causal models are equivalent in terms of interventional distribution, i.e., cannot be identified through observational or interventional data. Relative theories mostly discuss such causal consistency properties for some given categories of transformations, while we provide a novel transformation named *SCM reduction* for the latent variable identification problem.

## 3 PRELIMINARY

The identifiability problem of latent variables in latent causal model is described as follows: given a set of observed variables $\mathbf{v} = \{\mathbf{v}_i \in \mathbb{R}^{d_{v_i}}\}_{i=1}^n$ which are generated by a set of unknown latent variables $\mathbf{s} = \{\mathbf{s}_i \in \mathbb{R}^{d_{s_i}}\}_{i=1}^m$ and an unknown function $g$ such that $\mathbf{v} = g(\mathbf{s})$, under some given assumptions, whether the solution of each $\mathbf{s}_i$ can be determined uniquely up to some equivalence relations. In this paper, we define the equivalence relation by "equivalent information", i.e. two variables can predict each other precisely. Such equivalence is also called "equivalent up to invertible transformations" in other literatures (Kong et al., 2024; Lachapelle et al., 2024). A formal definition is given as follows.

**Definition 3.1** (*Subvariable and proper subvariable*). *Random variable* $\mathbf{u} \in \mathcal{U} \subseteq \mathbb{R}^m$ *is a subvariable of random variable* $\mathbf{v} \in \mathcal{V} \subseteq \mathbb{R}^n$, *iff. there exists a surjective mapping* $f : \mathcal{V} \to \mathcal{U}$ *such that* $\mathbf{u} = f(\mathbf{v})$ *for any* $(\mathbf{u}, \mathbf{v}) \in \mathrm{supp}(\mathbf{u}, \mathbf{v})$, *denoted as* $\mathbf{u} \preceq \mathbf{v}$. *Further,* $\mathbf{u}$ *is a proper subvariable of* $\mathbf{v}$ *iff.* $\mathbf{u} \preceq \mathbf{v}$ *and* $\mathbf{v} \npreceq \mathbf{u}$, *denoted as* $\mathbf{u} \prec \mathbf{v}$.

**Definition 3.2** (*Equivalent variable*). *Random variable* $\mathbf{u}$ *is equivalent to random variable* $\mathbf{v}$, *iff.* $\mathbf{u} \preceq \mathbf{v}$ *and* $\mathbf{v} \preceq \mathbf{u}$, *denoted as* $\mathbf{u} \sim \mathbf{v}$.

We aim to establish identifiablity theory for a wide range of models. As such, we consider structural causal models (SCM) (Pearl, 2009) as the underlying model, which has sufficient capability to explain the generation process of common variables. An SCM is a tuple $(G, F, P)$, where $G = (V, E)$ is the causal graph which is a directed acyclic graph with vertex set $V$ and edge set $E$. The vertex set $V$ consists of two disjoint sets, i.e. $V = U \cup S$, where $U = \{\mathbf{u}_i\}_{i=1}^m$ is a set of exogenous variables which are mutually independent, $S = \{\mathbf{s}_i\}_{i=1}^m$ is a set of endogenous variables. The edge set $E$ satisfies the following restrictions: there is no edge ends with any $\mathbf{u}_i$, and for any endogenous variable, there exists a path ends with it while starting from an exogenous variable. $F$ is a set of structural equations $\mathbf{s}_i = f_i(Pa(\mathbf{s}_i))$, where $f_i$ is a function and $Pa(\mathbf{s}_i)$ are the parent variables of $\mathbf{s}_i$. $P = p(\mathbf{u}_1, \cdots, \mathbf{u}_m) = \prod_{i=1}^m p(\mathbf{u}_i)$ is a probability distribution defined over all exogenous variables.

We further assume there are no directed path between observed variables, similar settings have been adopted by most works on latent variable identifiability (Khemakhem et al., 2020; Zheng et al., 2022; Kong et al., 2024). Since identifying all latent variables in an SCM is generally impossible without strong assumptions, we first reduce the original SCM into an equivalent SCM whose structure is a powerset bipartite graph (Section 4), and then discuss the identifiablity result (Section 5).

**Notations.** We use square brackets to denote the dimensional concatenation of variables, including: using $[\mathbf{x}, \mathbf{y}]$ to represent the concatenation of variables $\mathbf{x}, \mathbf{y}$; and using $[S]$ as the concatenation of all elements in variable set $S$.

## 4 REDUCTION OF STRUCTURAL CAUSAL MODELS

In this section, we discuss how to convert a structural causal model into a ideal form such that all latent variables are able to be identified.

Given an SCM $(G, F, P)$ as well as a set of observed variables $O = \{\mathbf{o}_l = \mathbf{s}_{i_l}\}_{l=1}^n$ which is a subset of the endogeous variables $S$, we would like to know which variables are most likely to be identified: the exogenous variables are naturally independent, and their information are limited to only go into their corresponding descendants, such restrictions give them the potential to be identified according to the graph structure; however for unobserved endogenous variables, they can be almost any forms of mixture of their ancestors in terms of information, such freedom makes them less likely to be identified. Based on such intuition, we focus on the identification of exogenous variables, which already provide sufficient information to generate the observed variables. However, some of the exogenous variables shares the same topology in the view of observed variables, i.e., have the same set of observed descendants. Without further observation, they are exchangeable and unlikely to be separated. Therefore, we merge such variables by dimensional concatenation, and expect their union can be identified.

Above analysis provides a motivation to convert a general SCM into an equivalent but reduced form. By saying "equivalent" here, we mean they share the same marginal distribution on observed variables, as well as the same marginal distribution on the (concatenated) exogenous variables. The word "reduced" means the converted SCM entails less vertices on the graph. We now formally define the reduction procedure which constructs a reduced form of the SCM $(G', F', P')$ w.r.t. observed variable set $O$ (see Fig. 1 for illustration):

**SCM Reduction.** Cluster all $\mathbf{u}_i$ by the same observed descendant set $\mathrm{OD}(\mathbf{u}_i) = \{\mathbf{o}_l \in O | \mathbf{u}_i \in \mathrm{An}(\mathbf{o}_l)\}$, where $\mathrm{An}(\mathbf{o}_l)$ is the ancestor set of $\mathbf{o}_l$. Remove the cluster with no observed descendants. Name the concatenation of a cluster $\mathbf{u}'_j$ with index $j = \sum_{l: \mathbf{o}_l \in \mathrm{OD}(\mathbf{u}_i)} 2^l$, whose binary representation indicates the topology w.r.t. $O$. Denote the set of all such index as $I$, and denote the elements in the cluster $\mathbf{u}'_j$ as $U_j$. For the vertex set $V' = U' \cup S'$, set $U' = \{\mathbf{u}'_j | j \in I\}$ and $S' = O$. The edge set $E'$ consists of all $\mathbf{u}'_j \to \mathbf{o}_l$ such that $j \& 2^l \neq 0$, where "&" is the bitwise "and" operation. For the structural equations $F'$, only include the equations for observed variables, i.e. $\mathbf{o}_l = f'_l(\{\mathbf{u}'_j | j \& 2^l \neq 0\}) = f_{i_l}(\mathrm{Pa}(\mathbf{s}_{i_l}))$ for $l$ in $\{1, \cdots, n\}$. For the probability distribution $P'$, set $p'(U') = \prod_{j \in I} p'(\mathbf{u}'_j) = \prod_{j \in I} \prod_{\mathbf{u}_i \in U_j} p(\mathbf{u}_i)$.

The above reduction process tries to cluster analogous exogenous variables and remove unobserved endogenous variables, which can be regarded as a specifically designed *exact transformation* as in causal abstraction literatures (Rubenstein et al., 2017). After reduction, the causal graph has been transformed uniquely into a special kind of bipartite graph, which we name *powerset bipartite graph*, or *PBG* for short. The definition is as follows (see Fig 2 for examples):

**Definition 4.1** *(Powerset bipartite graph). A directed graph $G = (V, E)$ is a powerset bipartite graph with size $n$, iff. it satisfies the following properties:*

1. *The vertex set $V$ consists of two disjoint sets $V_1$ and $V_2$, such that $V_1 \cup V_2 = V$, $V_1 \cap V_2 = \emptyset$.*

2. *$V_1$ consists of $n$ vertices, define an auxiliary set $N = \{1, 2, \cdots, n\}$ and establish a bijective mapping $\phi : V_1 \to N$.*

3. *$V_2$ consists of $m \leq 2^n - 1$ vertices, and there exists a set $M \subset \mathcal{P}(N) \backslash \{\emptyset\}$ and a bijective mapping $\psi : V_2 \to M$, where $\mathcal{P}(N)$ is the powerset of $N$, i.e. consisting all subsets of $N$.*

4. *There exists an edge from vertex $s$ to $t$ in the edge set $E$, iff. $s \in V_2$, $t \in V_1$, and $\phi(t) \in \psi(s)$.*

*Further, if $m = 2^n - 1$, i.e. $M = \mathcal{P}(N) \backslash \{\emptyset\}$, we call it a complete powerset bipartite graph.*

We call a structural causal model $(G, F, P)$ *PBG-SCM*, if $G$ is a powerset bipartite graph. In the following sections, we will then focus on the identifiability results for PBG-SCMs. Note that, if latent variables in a PBG-SCM can be identified, then the concatenated (according to topology) exogenous variables in original SCM can also be identified.

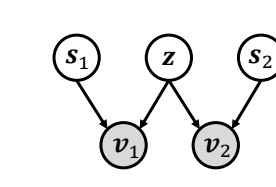 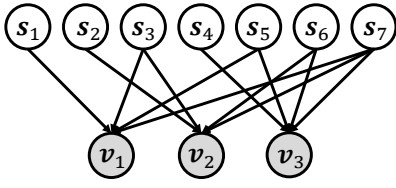

(a) Basis model.  (b) General PBG-SCM (size 3).

Figure 2: The models to be identified in Section5, which are structural causal models with complete PBG structures.

## 5 IDENTIFIABILITY OF LATENT VARIABLES IN PBG-SCMS

In this section, we first discuss the identifiability of a basis model (PBG-SCM with size 2), and then extend the results to general PBG-SCMs. We by default consider complete powerset bipartite graphs, since the missing nodes in a non-complete PBG-SCM can be viewed as constant latent variables in a complete PBG-SCM.

### 5.1 IDENTIFIABILITY RESULTS FOR BASIS MODEL

**Basis Model.** Consider a PBG-SCM (see Fig. 2a) with latent variables $\mathbf{s} = (\mathbf{s}_1, \mathbf{z}, \mathbf{s}_2) \in \mathcal{S}$ and observed variables $\mathbf{v} = (\mathbf{v}_1, \mathbf{v}_2) \in \mathcal{V}$, where $\mathbf{s}_1 \in \mathbb{R}^{d_{\mathbf{s}_1}}, \mathbf{z} \in \mathbb{R}^{d_z}, \mathbf{s}_2 \in \mathbb{R}^{d_{\mathbf{s}_2}}, \mathbf{v}_1 \in \mathbb{R}^{d_{\mathbf{v}_1}}, \mathbf{v}_2 \in \mathbb{R}^{d_{\mathbf{v}_2}}$. The corresponding structural equations are

$$\mathbf{v}_1 = g_1(\mathbf{z}, \mathbf{s}_1), \ \mathbf{v}_2 = g_2(\mathbf{z}, \mathbf{s}_2), \tag{1}$$

where $g_1$ and $g_2$ are the generation functions. We denote the probability distribution of observed variables as $p(\mathbf{v}_1, \mathbf{v}_2)$, and the generation model as $(p_{\mathbf{s}_1}, p_z, p_{\mathbf{s}_2}, g_1, g_2)$. According to Thm. 1, latent variables in a basis model are identifiable, if the following assumptions are satisfied:

**Assumption 1** *(Verifiable identifiability conditions for basis model).*

  *i [Invertibility] There exists a differentiable invertible function $g : \mathcal{S} \to \mathcal{V}$ such that $\mathbf{v} = g(\mathbf{s})$, and $g$ has a differentiable inverse $g^{-1}$.*

  *ii [Independence] Latent variables $\mathbf{s}_1, \mathbf{z}, \mathbf{s}_2$ are mutually independent, i.e., $p(\mathbf{s}_1, \mathbf{z}, \mathbf{s}_2) = p(\mathbf{s}_1)p(\mathbf{z})p(\mathbf{s}_2)$.*

  *iii [Minimality] The shared latent variable $\mathbf{z}$ is minimal, i.e., there does not exist a model $(p'_{\mathbf{s}_1}, p'_z, p'_{\mathbf{s}_2}, g'_1, g'_2)$ satisfying assumption $i$ and $ii$, such that $p'(\mathbf{v}'_1, \mathbf{v}'_2) = p(\mathbf{v}_1, \mathbf{v}_2)$ for any $(\mathbf{v}'_1, \mathbf{v}'_2) = (\mathbf{v}_1, \mathbf{v}_2) \in \mathcal{V}$ and $\mathbf{z}' \prec \mathbf{z}$.*

**Theorem 1** *(Basis model identifiability) For two generation models $(p_{\mathbf{s}_1}, p_z, p_{\mathbf{s}_2}, g_1, g_2)$ and $(\hat{p}_{\mathbf{s}_1}, \hat{p}_z, \hat{p}_{\mathbf{s}_2}, \hat{g}_1, \hat{g}_2)$ satisfying the data generation process in Equation 1, if both models satisfy Assumption 1 and $p(\mathbf{v}_1, \mathbf{v}_2) = \hat{p}(\hat{\mathbf{v}}_1, \hat{\mathbf{v}}_2)$ for any $(\mathbf{v}_1, \mathbf{v}_2) = (\hat{\mathbf{v}}_1, \hat{\mathbf{v}}_2) \in \mathcal{V}$, then there is $\mathbf{s}_1 \sim \hat{\mathbf{s}}_1, \mathbf{z} \sim \hat{\mathbf{z}}, \mathbf{s}_2 \sim \hat{\mathbf{s}}_2$.*

**Remark.** The invertibility condition is a basic requirement in almost all existing identifiabilty theories, without which the latent variables cannot be recovered from observed variables. The independence condition is also widely applied in related works, without which the information in latent variables can be mixed and cannot be separated. It is also a natural corollary from the reduction process of SCM, since latent variables in a PBG-SCM are dimensional concatenations of disjoint and independent exogenous variables. The minimality condition is a key contribution of our identifiability theory, which requires the shared latent variable $\mathbf{z}$ to have minimal information. This prevents $\mathbf{z}$ from plundering information from $\mathbf{s}_1$ or $\mathbf{s}_2$. Though simple in its concept, this assumption is overlooked by existing literatures. We then discuss what this assumption really conveys and why it fails to gain much attention.

**Definition 5.1** *(Intrinsic dimension). The intrinsic dimension of a random variables $\mathbf{x} \in \mathbb{R}^{d_x}$ is the minimal dimension among its equivalent variables, denoted as $\mathrm{IDim}(\mathbf{x}) = \min\{\dim(\mathbf{z}) | \mathbf{z} \sim \mathbf{x}\}$.*

**Proposition 5.1** *(Violation of minimality) For a basis model $\mathcal{M} = (p_{s_1}, p_z, p_{s_2}, g_1, g_2)$, if it satisfies assumption $i$ and $ii$ but does not satisfy assumption $iii$, then there exists an identifiable model $\mathcal{M}' = (p'_{s_1}, p'_z, p'_{s_2}, g'_1, g'_2)$ satisfying Assumption 1, such that $p'(\mathbf{v}'_1 = \mathbf{v}_1, \mathbf{v}'_2 = \mathbf{v}_2) \equiv p(\mathbf{v}_1, \mathbf{v}_2)$, and there exist random variables $\mathbf{z}_0, \mathbf{z}_1, \mathbf{z}_2$ such that $\mathbf{z} \sim [\mathbf{z}_0, \mathbf{z}_1, \mathbf{z}_2]$, $\mathrm{IDim}(\mathbf{z}_0) < \mathrm{IDim}(\mathbf{z})$, and $\mathbf{s}'_1 \sim [\mathbf{s}_1, \mathbf{z}_1]$, $\mathbf{z}' \sim \mathbf{z}_0$, $\mathbf{s}'_2 \sim [\mathbf{s}_2, \mathbf{z}_2]$.*

The concept "intrinsic dimension" defined above is not new (Camastra & Staiano, 2016; Pope et al., 2020), which describes the "dimension of the necessary information" of a variable. According to Proposition 5.1, a non-minimal $\mathbf{z}$ has oversized intrinsic dimension, and the extra information are dimensional and indeed from $\mathbf{s}_1$ or $\mathbf{s}_2$. According to Corollary 5.1, by restricting the dimension of learned $\hat{\mathbf{z}}$, we are also able to recover the ground truth latent variables. This means, if we know the intrinsic dimension of latent variables in advance, and ensure that invertibility and independence are satisfied, then the minimality condition will automatically be satisfied by setting latent dimensions the same as ground truth. This might provide an explanation why minimality condition is overlooked, since "known dimension" is a default setting in experiment design of most literatures. As a result, we argue that the minimality condition is important, and researchers should consider experiment settings with unknown latent dimension for validating their identifiability results.

**Corollary 5.1** *(Substitute for minimality) For a basis model $\mathcal{M} = (p_{s_1}, p_z, p_{s_2}, g_1, g_2)$ satisfying Assumption 1 and another basis model $\hat{\mathcal{M}} = (\hat{p}_{s_1}, \hat{p}_z, \hat{p}_{s_2}, \hat{g}_1, \hat{g}_2)$ satisfying assumption $i$ and $ii$, and $p(\mathbf{v}_1, \mathbf{v}_2) = \hat{p}(\hat{\mathbf{v}}_1, \hat{\mathbf{v}}_2)$ for any $(\mathbf{v}_1, \mathbf{v}_2) = (\hat{\mathbf{v}}_1, \hat{\mathbf{v}}_2) \in \mathcal{V}$, if $\dim(\hat{\mathbf{z}}) = \mathrm{IDim}(\mathbf{z})$, then $\hat{\mathcal{M}}$ also satisfies assumption $iii$, thus there is $\mathbf{s}_1 \sim \hat{\mathbf{s}}_1, \mathbf{z} \sim \hat{\mathbf{z}}, \mathbf{s}_2 \sim \hat{\mathbf{s}}_2$.*

### 5.2 IDENTIFIABILITY RESULTS FOR GENERAL MODEL

**General Model**. Consider an complete PBG-SCM (see Fig. 2b for an example) with latent variables $\mathbf{s} = (\mathbf{s}_1, \cdots, \mathbf{s}_m) \in \mathcal{S}$ and observed variables $\mathbf{v} = (\mathbf{v}_1, \cdots, \mathbf{v}_n) \in \mathcal{V}$, where $\mathbf{s}_i \in \mathbb{R}^{d_{\mathbf{s}_i}}, \mathbf{v}_j \in \mathbb{R}^{d_{\mathbf{v}_j}}$ for any $i$ and $j$. Note that there is $m = 2^n - 1$ for a complete PBG-SCM, and we allow for constant latent variables which leads to a non-complete PBG-SCM. The corresponding structural equations are

$$\mathbf{v}_j = g_j(\{\mathbf{s}_i | i \& 2^j \neq 0\}), \tag{2}$$

where $g_j$ are the generation functions, "$\&$" is the bitwise "and" operation. Eq. 2 means the connection between $\mathbf{v}_j$ and $\mathbf{s}_i$ exists iff. the $j$-th digit (from right to left) in the binary representation of $i$ is 1. We denote the probability distribution of observed variables as $p(\mathbf{v})$, and the generation model as $(p_s, \{g_j\}_{j=1}^n)$. According to Thm 2, latent variables in a PBG-SCM are identifiable, if the following assumptions are satisfied:

**Assumption 2** *(Verifiable identifiability conditions for general model).*

> *i [Invertibility] There exists a differentiable invertible function $g : \mathcal{S} \to \mathcal{V}$ such that $\mathbf{v} = g(\mathbf{s})$, and $g$ has a differentiable inverse $g^{-1}$.*

> *ii [Independence] Latent variables $\{\mathbf{s}_i\}_{i=1}^m$ are mutually independent, i.e., $p(\mathbf{s}) = \prod_{i=1}^m p(\mathbf{s}_i)$.*

> *iii [Hierarchical minimality] Any latent variable $\mathbf{s}_i$ with at least 2 descendants is minimal, i.e., there does not exist a model $(p'_s, \{g'_j\}_{j=1}^n)$ satisfying assumption $i$ and $ii$, such that $p'(\mathbf{v}') = p(\mathbf{v})$ for any $\mathbf{v}' = \mathbf{v} \in \mathcal{V}$ and $\mathbf{s}'_i \prec \mathbf{s}_i, \mathbf{s}'_k \sim \mathbf{s}_k$ for any $k \in \{k | k \neq i, k \& i = i\}$.*

**Theorem 2** *(General model identifiability) For two PBG-SCMs $(p_s, \{g_j\}_{j=1}^n)$ and $(\hat{p}_s, \{\hat{g}_j\}_{j=1}^n)$ satisfying the data generation process in Equation 2, if both models satisfy Assumption 2 and $p(\mathbf{v}) = \hat{p}(\hat{\mathbf{v}})$ for any $\hat{\mathbf{v}} = \mathbf{v} \in \mathcal{V}$, then there is $\mathbf{s}_i \sim \hat{\mathbf{s}}_i$ for any $i$.*

**Sketch of proof.** A general PBG-SCM can be identified by iteratively applying basis models on (concatenations of) observed variables and intermediate variables. The main process include 2 steps: 1. We use one observed variable as $\mathbf{v}_1$ and concatenation of the rest as $\mathbf{v}_2$, build basis models to get a series of intermediate variable, which are equivalent to concatenation of some latent variables in PBG-SCM. 2. We further apply basis models iteratively on intermediate variables by some rules, till each single latent variable is separated from the concatenations. An illustrative example can be found in Fig. 4b, and detailed constructive proof can be found in Appendix A.7.

**Remark.** The invertibility and independence conditions are similar to that in basis model, we mainly focus on the hierarchical minimality condition. It requires that for a latent variable shared by multiple descendants, it should be minimal once their "upper variables" are already minimal. By saying "upper variable" here, we mean other latent variables whose descendant set is a superset. An intuitive explanation is, for information of individual observed variable (or smaller groups of observed variables), it should not be put into a shared latent variable (or latent variable corresponding to larger group). If this assumption is violated, an "upper" variable may plunder arbitrary information from "lower" variables, thus is unlikely to be identified.

Combining all above conclusions, we extend the identifiability results to the original SCM. Given a set of observed variables which are generated by an SCM with no directed path among observed variables, as long as the function between exogenous variables and observed variables is differentiable and globally invertible, then we can guarantee a unique identification result (up to invertible transformation). Further, if shared latent variables in original SCM entail minimal information, then we can guarantee the identification of (concatenations of) original exogenous variables in the finest grain. Such assumptions are quite mild, which ensures the broad applicability of our theory.

# 6 EXPERIMENTS

In this section, we demonstrate through experiments that, for an identifiable PBM-SCM, learning mechanisms corresponding to each assumptions should be considered in algorithm design for successful identification. We first introduce the datasets and metrics used in our experiments, and then show the identification results for basis model and general PBM-SCM, respectively.

## 6.1 DATASETS AND METRICS

**Synthetic datasets for basis model.** For the grouth truth model, we sample $\mathbf{s}_1, \mathbf{z}, \mathbf{s}_2$ from standard normal distribution $\mathcal{N}(\mathbf{0}, \mathbf{I})$ (with differnt dimensions), so that they are mutually independent. We by default set the dimensions of latent variables as $d_{s_1} = 3, d_z = 5, d_{s_2} = 4$. For the observed variables $\mathbf{v}_1, \mathbf{v}_2$, we set $d_{v_1} = d_{v_2} = 10$, and use three different mechanisms for the generation process, which therefore constitutes different datasets:

- *Concatenation* dataset. Apply concatenation in dimension and then do invertible transformation for latent variables, i.e. $\mathbf{v}_1 = f_1([\mathbf{s}_1, \mathbf{z}]), \mathbf{v}_2 = f_2([\mathbf{z}, \mathbf{s}_2])$.

- *Split* dataset. Split the space of $\mathbf{z}$ into subspaces before concatenation, and then do invertible transformations, i.e. $\mathbf{z}^+ = \max(\mathbf{z}, 0), \mathbf{z}^- = \min(\mathbf{z}, 0), \mathbf{v}_1 = f_1([\mathbf{s}_1, \mathbf{z}^+]), \mathbf{v}_2 = f_2([\mathbf{z}^-, \mathbf{s}_2])$.

- *Fusion* dataset. Fuse the information of $\mathbf{z}$ deeply in one side, and then do invertible transformations, i.e. $\mathbf{v}_1 = f_1(\mathbf{s}_1) + f_2(\mathbf{z}), \mathbf{v}_2 = f_3([\mathbf{z}, \mathbf{s}_2])$.

For the transformation function $f_i$, we use randomly-initialized multi-layer perceptrons (MLP) with *Tanh* activations. Detailed configuration of such MLP can be found in Appendix A.8. We checked the rank of weight matrices in each linear layer to ensure they are of full rank, therefore any $f_i$ is guaranteed to be invertible. The invertibility of whole generation process of all above models are easy to verify, note that the *Split* and *Fusion* datasets are globally invertible but not locally invertible, i.e. $\mathbf{s}_1$ and $\mathbf{z}$ cannot be recovered given only $\mathbf{v}_1$. The minimality condition is also satisfied by all above models, since moving any dimension in $\mathbf{z}$ to $\mathbf{s}_1$ (or $\mathbf{s}_2$) will make $\mathbf{v}_2$ (or $\mathbf{v}_1$) unrecoverable.

**Synthetic dataset for general model.** Similar to the construction of basis model, we use a complete PBG-SCM with size 3, and also sample $\mathbf{s}_i$ from standard normal distribution $\mathcal{N}(\mathbf{0}, \mathbf{I})$ for any $i \in \{1, \cdots, 7\}$. We set the dimension of all latent variables as $d_{s_i} = 2$ and the dimension of all observed variables as $d_{v_j} = 10$. We generate the observed variables by $\mathbf{v}_j = f_j([\mathbf{s}_i | i \& 2^j \neq 0]), j \in \{1, 2, 3\}$, where $f_j$ is also an invertible function implemented by MLP. The invertibility and minimality of this PBG-SCM can be similarly verified as the above basis models.

**Metrics.** We follow the work of (Kong et al., 2024) to use *coefficient of determination* ($R^2$ score) for evaluation of the degree of variable equivalence or variable independence. $R^2$ score takes value in $[0, 1]$, and larger $R^2$ represents better identification result. The $R^2$ score for equivalence takes value 1 for equivalent variables, while $R^2$ score for independence takes value 1 for independent variables. Detailed definition can be found in Appendix A.8.

Table 1: The $R^2$ scores ($\times 100$) of basis model identification algorithms under 3 different datasets. Numbers are the mean value over 5 runs with different random seeds. Number in the brackets is the standard error of the mean.

| Methods | Concatenation | | Split | | Fusion | |
|---|---|---|---|---|---|---|
| | $\mathbf{s}_1$/$\mathbf{z}$/$\mathbf{s}_2$ | Avg. | $\mathbf{s}_1$/$\mathbf{z}$/$\mathbf{s}_2$ | Avg. | $\mathbf{s}_1$/$\mathbf{z}$/$\mathbf{s}_2$ | Avg. |
| AE | 49.3/88.8/66.0 | 68.1 | 52.5/69.2/66.2 | 62.6 | 50.4/83.7/59.9 | 64.7 |
| ($d_z = 7$) | ($\pm$1.9/0.7/2.8) | ($\pm$1.0) | ($\pm$2.7/2.4/2.8) | ($\pm$1.9) | ($\pm$4.0/1.3/1.6) | ($\pm$1.7) |
| AE+CLUB | 76.8/90.1/67.8 | 82.7 | 68.7/62.2/67.0 | 66.0 | 74.7/88.8/83.0 | 82.2 |
| ($d_z = 7$) | ($\pm$8.0/3.1/7.0) | ($\pm$3.7) | ($\pm$4.5/3.2/5.2) | ($\pm$3.4) | ($\pm$4.9/0.6/3.0) | ($\pm$1.0) |
| AE | 67.8/**99.5**/78.3 | 81.8 | 64.8/**97.8**/74.7 | 79.1 | 73.1/**99.5**/72.5 | 81.7 |
| ($d_z = 5$) | ($\pm$1.1/0.0/1.7) | ($\pm$0.6) | ($\pm$3.4/0.1/1.4) | ($\pm$1.5) | ($\pm$1.4/0.0/1.7) | ($\pm$0.7) |
| AE+CLUB | **96.6**/**99.5**/**97.2** | **97.8** | **92.3**/95.7/**93.7** | **93.9** | **97.3**/99.4/**97.0** | **97.9** |
| ($d_z = 5$) | ($\pm$0.1/0.0/0.0) | ($\pm$0.0) | ($\pm$1.4/1.2/1.6) | ($\pm$1.3) | ($\pm$1.4/0.1/0.1) | ($\pm$0.6) |

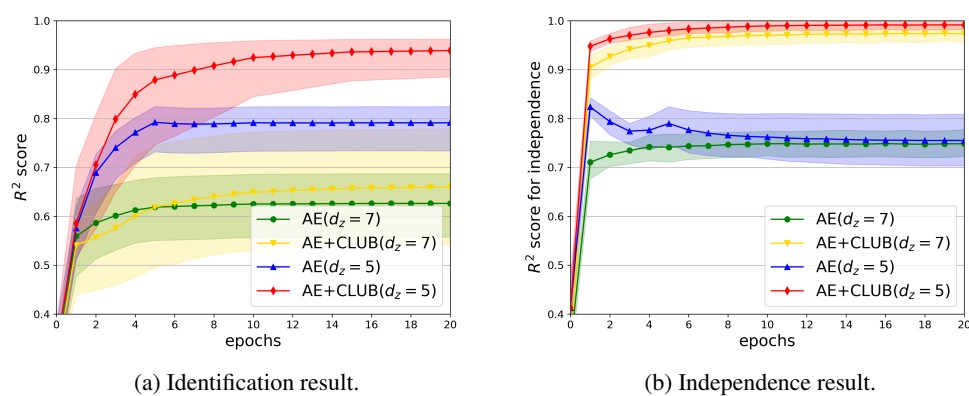

(a) Identification result.          (b) Independence result.

Figure 3: The mean of averaged $R^2$ score curve during training of identification algorithms on the *Split* dataset. Shaded area represents the region from min value to max value over 5 runs.

## 6.2 BASIS MODEL IDENTIFICATION

For a basis model, our goal is to identify all latent variables $\mathbf{s}$ given observations on $\mathbf{v}$. Thm. 1 tells us that our learned model should satisfy Assumption 1 in order to learn equivalent latent variables. As such, we apply an autoencoder (AE) framework for the invertibility condition, which encodes $\mathbf{v}$ into $\hat{\mathbf{s}}$ by an MLP and then decode $\hat{\mathbf{s}}$ back to $\mathbf{v}$ by another MLP. An MSE loss is adopted to ensure invertibility. For the independence condition, we apply Contrastive Log-ratio Upper Bound (CLUB) (Cheng et al., 2020) method as extra loss between each latent variable and the rest, which is a widely used approach for guarantee of independence. For the minimality condition, we set the dimension of $\mathbf{z}$ in our model by $d_z = 5$, which equals the intrinsic dimension of ground truth, thus satisfies the conditions in Corollary 5.1. For a comparison, we set $d_z = 7$ as scenarios that minimality condition is not satisfied.

Tab. 1 shows the identification result of different learning strategies. We can see that if all conditions are satisfied (AE+CLUB, $d_z = 5$), the learned latent variables are almost equivalent to that in ground truth model, indicated by $R^2 > 0.9$. Such results are consistent among all 3 datasets, showing that latent variables in basis model can be identified if Assumption 1 is satisfied. Either independence condition is violated (AE, $d_z = 5$) or minimality condition is violated (AE+CLUB, $d_z = 7$), $R^2$ scores decreases severely, indicating a non-perfect identification. The setting of AE($d_z = 5$) can be viewed as a baseline, which is the method used by default in existing identification works. We can see that, though such methods achieve considerable $R^2$ scores (around $0.8$), it in fact does not indicate successful identification by comparing with best $R^2$ scores. Fig. 3 further shows the details during training. We can see that the training process is relatively stable for AE+CLUB($d_z = 5$), and the CLUB loss does ensure independence among latent variables.

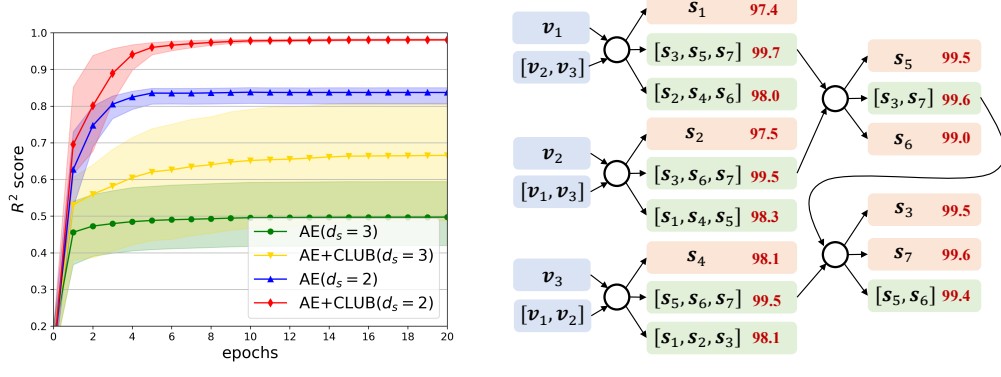

(a) Identification by a single autoencoder.      (b) Identification by basis models ($R^2 \times 100$).

Figure 4: The $R^2$ scores for identification results on general PBG-SCM. (a) Results of an implementation similar to that of a single basis model. Shaded area represents the region from min value to max value over 5 runs. (b) Results of the implementation by iteratively building 5 different basis models (hollow circles). $R^2$ score is shown next to the corresponding variable.

## 6.3 GENERAL PBG-SCM IDENTIFICATION

For general PBG-SCMs, we design 2 different identification algorithms. One is similar to that for basis model, which is also an autoencoder framework. We set $d_s = 2$ for each $\mathbf{s}_i$ to satisfy the minimality condition, and $d_s = 3$ for scenarios that minimality condition is violated. The other is an optimized implementation of the constructive proof for Thm 2 in Section 5.2, which iteratively apply 5 different basis models on oberserved variables or identified intermediate variables (see Fig. 4b).

Fig. 4a shows the identification results of a single autoencoder model. Again we can see that AE+CLUB($d_s = 2$) achieves almost optimal identification result with $R^2$ score close to $1.0$, and is quite stable in multiple runs, indicating that latent variables in a general PBG-SCM can be identified if Assumption 2 is satisfied. Any violation of independence or minimality condition will lead to significantly worse identification, as indicated by the gap in $R^2$ compared with other curves. Fig. 4b provide the identification results of the iterative approach with basis model. We can see that the intermediate latent variables are precisely identified in each basis model, and finally all single latent variables in PBG-SCM are identified with high $R^2$ scores. These results provide support for the constructive proof of Thm. 2.

## 7 CONCLUSION

We proposed a novel identifiability theory for latent variable identification in nonlinear causal models with single-domain data. Our theory guarantees that, for any structural causal models which no directed path among observed variables, there exists a unique PBG-SCM which is equivalent in terms of joint distribution of all variables, whose latent variables are identifiable if invertibility, independence, and minimality conditions are satisfied. Our theory provides guidance for the design of identification algorithm, in a sense that each conditions should be considered for successful identification.

**Limitations.** The major limitations of this work include: 1. We follow existing works and assume there are no directed path among observed variables in an SCM, and all variables are continuous. Such assumptions may not hold true in specific scenarios. Future works may consider an extension to more general SCMs, allowing for the existence of directed paths and discrete variables. 2. This work is mainly a theoretical work, thus has limited contribution in algorithm design. Like existing works, the succeeded algorithms in our experiments still need pre-known knowledge of the intrinsic dimension of latent variables. Designing practical identification algorithms according to the theory is required in future works.

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

# A  APPENDIX

## A.1  FURTHER DISCUSSIONS ON SCM REDUCTION

We explain the numbering rule in our SCM Reduction procedure, which is also used in our general model. See Fig. 5 for an example. For a concatenated cluster $\mathbf{u}'_{19}$, the binary representation of 19 is $[10101]_2 = [00001]_2 + [00100]_2 + [10000]_2 = 2^1 + 2^3 + 2^5$, so that the observed descendants of this cluster are $\mathbf{o}_1$, $\mathbf{o}_3$, and $\mathbf{o}_5$. Under this rule, the equation $j\&2^i \neq 0$ has the following meaning: $\mathbf{o}_i$ is a descendant of $\mathbf{u}'_j$.

We then give a proof here for our claim that, any SCM $(G, F, P)$ can be transformed into a PBG-SCM by the reduction process given a subset of endogenous variables as observed set $O$, if there is no directed paths among variables in $O$.

The reduction process has provided a construction procedure of a new SCM. Since the structural equations for each observed variables are essentially the unfolding form of those in original SCM, so the observational distribution remains unchanged. We only need to prove that the transformed graph is a PBG.

Consider the 4 properties of PBG in Def. 4.1. We discuss them one by one.

1. We set $V_1 = S' = O$, $V_2 = U'$. Since $V = U' \cup S'$, so that $V_1 \cup V_2 = V$. Since variables in $O$ are endogenous variables, and variables in $U'$ are concatenations of exogenous variables, so that $V_1 \cap V_2 = \emptyset$.

2. We define $\phi$ as the following: for each $\mathbf{o}_l \in V_1$, map it to $l$. Clearly, $\phi$ is a bijective mapping.

3. We define $\psi$ as the following: for each $\mathbf{u}'_j \in V_2$, map it to the index set $\{l|j\&2^l \neq 0\}$. Define set $M$ as $M = \{\psi(\mathbf{u}'_j)|\mathbf{u}'_j \in V_2\}$. Note that for each observed descendant $\mathbf{o}_l$ of $\mathbf{u}'_j$, there is $j\&2^l \neq 0$ according to numbering rule that $j = \sum_{l:\mathbf{o}_l \in \mathrm{OD}(\mathbf{u}_i)} 2^l$. Consequently, $\psi(\mathbf{u}'_j)$ are the indices of the observed descendants for $\mathbf{u}'_j$. Any $\psi(\mathbf{u}'_j)$ is not empty, since the cluster with no observed descendants has been removed during reduction. $\psi(\mathbf{u}'_i) \neq \psi(\mathbf{u}'_j)$ if $i \neq j$, since variables with the same observed descendant set have been clustered and merged, so that $\psi$ is bijective. Now there is $M \subset \mathcal{P}(N)\backslash\{\emptyset\}$. Again, according to numbering rule that $j = \sum_{l:\mathbf{o}_l \in \mathrm{OD}(\mathbf{u}_i)} 2^l$, there is $0 < j \leq 2^n - 1$, so that $m \leq 2^n - 1$.

4. According to the reduction procedure, the edge set $E'$ consists of all $\mathbf{u}'_j \to \mathbf{o}_l$ such that $j\&2^l \neq 0$, which means there exists and only exists edges for a vertex $s = \mathbf{u}'_j \in V_2$ to another vertex $t = \mathbf{o}_l \in V_1$, satisfying $j\&2^l \neq 0$. So that $\phi(t) = \phi(\mathbf{o}_l) = l \in \{l|j\&2^l \neq 0\} = \psi(\mathbf{u}'_j) = \psi(s)$.

We can see that all properties of PBG are satisfied by the reduced graph, so that it is a PBG.

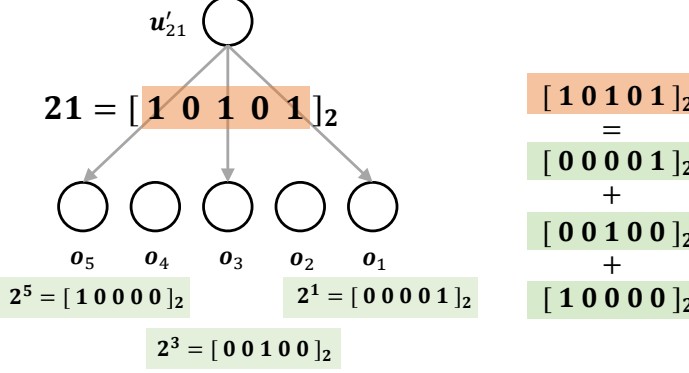

Figure 5: An example of the numbering rule in our SCM Reduction.

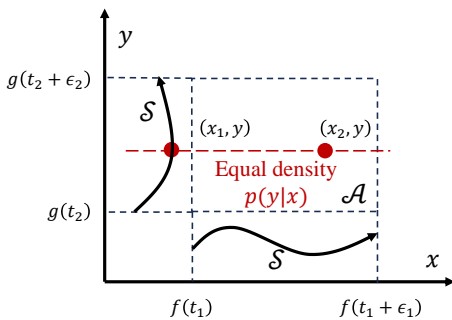

Figure 6: The area considered in Lemma A.1.

### A.2  Lemma for Thm. 1

**Lemma A.1** *Given a one-dimensional random variable $t$ with probability density $p(t)$ whose support $supp(t)$ is a 1-dimensional manifold, if there exist differentiable function $f, g : \mathbb{R} \to \mathbb{R}$ and $f, g \in \mathcal{C}^1$ such that $x = f(t)$ and $y = g(t)$, and there exist $t_1, t_2 \in \mathrm{supp}(t)$ such that $\frac{\mathrm{d}x}{\mathrm{d}t}|_{t=t_1} \neq 0$ and $\frac{\mathrm{d}y}{\mathrm{d}t}|_{t=t_2} \neq 0$, then there is $x \not\perp y$.*

**Proof.** We assume $\frac{\mathrm{d}x}{\mathrm{d}t}|_{t=t_1} > 0$ w.l.o.g.. Consider a neighbourhood region of $t_1$, i.e. $U_1 = [t_1, t_1 + \epsilon_1]$, since $f$ has continuous first-order derivative and $supp(t)$ is a 1-d manifold, there exists $\epsilon_1 > 0$ such that $\forall t \in U_1$, $\frac{\mathrm{d}x}{\mathrm{d}t} > 0, p(t) > 0$. Similarly, We assume $\frac{\mathrm{d}y}{\mathrm{d}t}|_{t=t_2} > 0$, and $\forall t \in U_2$, $\frac{\mathrm{d}y}{\mathrm{d}t} > 0, p(t) > 0$, where $U_2 = [t_2, t_2 + \epsilon_2]$. Consider the parametric equations of $x$ and $y$, the graph of which would be a continuous curve on the 2d $x$-$y$ plane, we denote it as $\mathcal{S}$. Consider the rectangular area $x \in (f(t_1), f(t_1 + \epsilon_1))$, $y \in (g(t_2), g(t_2 + \epsilon_2))$, denoted as $\mathcal{A}$ (see Fig. 6 for illustration). Consider the probability density $p(y|x)$ in $\mathcal{A}$, we know for all $x \in (f(t_1), f(t_1 + \epsilon_1))$, there is $p(x) > 0$. According to the continuous nature of curve $\mathcal{S}$ in region $U_2$, $y$ can take almost any value in $(g(t_2), g(t_2 + \epsilon_2))$, i.e., for almost all $y = y_0 \in (g(t_2), g(t_2 + \epsilon_2))$, there exists $x = x_0 \in \mathrm{supp}(x)$ such that $p(y_0|x_0) > 0$. If $x \perp y$, then there would be $p(y|x_1) = p(y|x_2)$ for any $x_1, x_2 \in \mathrm{supp}(x)$. So that there would be $p(y|x) > 0$ for almost everywhere in $\mathcal{A}$. That means, curve $\mathcal{S}$ is a space-filling curve with infinite length. However, since all points on $\mathcal{S}$ have positive density with positive infimum, this will lead to infinite total probability, thus causes contradiction. As a result, we have $x \not\perp y$. $\qquad\square$

**Remark.** The above proof may not be explanatory enough, we then give an intuitive explanation here (not a proof). Define $U_1$ the same as above. According to whether there exists $t \in U_1$ such that $\frac{\mathrm{d}y}{\mathrm{d}t} \neq 0$, we split the problem into two cases:

Case 1: such $t$ exists, we denote it as $t_3$ and assume $\frac{\mathrm{d}y}{\mathrm{d}t}|_{t=t_3} > 0$ w.l.o.g., we can similarly find $\epsilon > 0$ such that $\frac{\mathrm{d}y}{\mathrm{d}t} > 0$ for all $t$ in the region $U = [t_3, t_3 + \epsilon] \subset U_1$. This means $f$ and $g$ are monotonic and thus invertible in $U$. Denote the inverse of $f$ as $f^{-1}$, so that $y = g(f^{-1}(x))$, and $\forall x \in [f(t_3), f(t_3 + \epsilon)]$, $\frac{\mathrm{d}y}{\mathrm{d}x} = \frac{\mathrm{d}y}{\mathrm{d}t} \cdot \frac{\mathrm{d}t}{\mathrm{d}x} > 0$, indicating $x$ and $y$ are correlated.

Case 2: such $t$ does not exist, which means $\frac{\mathrm{d}y}{\mathrm{d}t} = 0$ iff. $\frac{\mathrm{d}x}{\mathrm{d}t} \neq 0$. This indicates a synchronous behavior of $x$ and $y$, i.e., $y$ changes as long as $x$ stops changing, and vice versa. As $f, g \in \mathcal{C}^1$, we can find a boundary $t = a$ such that $\frac{\mathrm{d}x}{\mathrm{d}t} = 0$ for $t \leq a$ and $\frac{\mathrm{d}x}{\mathrm{d}t} \neq 0$ for $t > a$ (or the contrary, see Fig. 7 for illustration). Consider a small region around $t = a$, e.g. $[a - \epsilon, a + \epsilon]$, denote $m = f(a)$, $n = g(a)$, then there is $p(x = m|y = n) = 0$ and $p(x = m|y \neq n) > 0$ in this region, indicating $x$ and $y$ are correlated.

### A.3  Proofs of Thm. 1

**Proof.** Before starting the detailed proof, we first transform each latent variable in both models $(p_{s_1}, p_z, p_{s_2}, g_1, g_2)$ and $(\hat{p}_{s_1}, \hat{p}_z, \hat{p}_{s_2}, \hat{g}_1, \hat{g}_2)$ (6 variables in total) into their "prime" counterpart according to the following procedure (take $\mathbf{z} \to \mathbf{z}'$ as example):

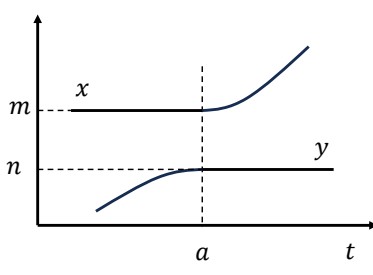

Figure 7: The synchronous behavior of $x$ and $y$ in Case 2 of explanations of Lemma A.1.

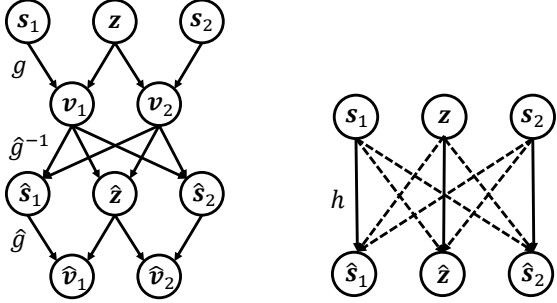

Figure 8: The data generation procedure (left) and corresponding edges of $J_h$ (right). Dashed lines are edges that are not expected to exist.

- Do nonlinear ICA over $\mathbf{z}$ by taking an invertible function $f_1$ to get $\mathbf{z}^{\perp} = f_1(\mathbf{z})$, after which all dimensions of $\mathbf{z}^{\perp}$ are independent. This is always feasible according to the existence of solution in non-linear ICA (Hyvärinen & Pajunen, 1999).

- Remove all constant dimensions in $\mathbf{z}^{\perp}$ to get the final variable $\mathbf{z}'$. Denote this operation as $\mathbf{z}' = f_2(\mathbf{z}^{\perp})$.

The above transformation $f = f_2 \circ f_1$ is invertible, so that $\mathbf{z} \sim \mathbf{z}'$. After the transformations, we get two "prime" models $(p'_{s_1}, p'_z, p'_{s_2}, g'_1, g'_2)$ and $(\hat{p}'_{s_1}, \hat{p}'_z, \hat{p}'_{s_2}, \hat{g}'_1, \hat{g}'_2)$, which still satisfies all assumptions. Note that the new structural equations can be constructed by applying the inverse of $f$ first, e.g., $g'_1(\mathbf{s}'_1, \mathbf{z}') = g_1(f_{s_1}^{-1}(\mathbf{s}'_1), f_z^{-1}(\mathbf{z}'))$. As long as we can prove the "prime" models are equivalent (in terms of $\sim$ over latent variables), then the original models are also equivalent, e.g., $\mathbf{z}' \sim \hat{\mathbf{z}}' \Rightarrow f_z(\mathbf{z}) \sim f_{\hat{z}}(\hat{\mathbf{z}}) \Rightarrow \mathbf{z} \sim \hat{\mathbf{z}}$.

After the above transformations, it is always feasible to change one dimension of the latent variables while keeping other dimensions unchanged, which is an important technique in the following proof. We then omit the "prime" symbols for simplicity, and start the main proof then.

To better distinguish the two models, we call $\mathbf{s}_1, \mathbf{z}, \mathbf{s}_2$ the ground truth latents and $\hat{\mathbf{s}}_1, \hat{\mathbf{z}}, \hat{\mathbf{s}}_2$ the learned latents. We use the notation $[\mathbf{x}, \mathbf{y}]$ to denote the concatenation of vector $\mathbf{x}$ and $\mathbf{y}$. We first summarize the data generation procedure for all variables (see Fig. 8 for illustration):

1. Generate observed variables by ground truth latents: $[\mathbf{v}_1, \mathbf{v}_2] = g([\mathbf{s}_1, \mathbf{z}, \mathbf{s}_2])$;

2. Encode observed variables into learned latents: $[\hat{\mathbf{s}}_1, \hat{\mathbf{z}}, \hat{\mathbf{s}}_2] = \hat{g}^{-1}([\mathbf{v}_1, \mathbf{v}_2])$, the existence of $\hat{g}^{-1}$ is guaranteed by assumption $i$ (invertibility), and the equivalence is guaranteed by $p(\mathbf{v}_1, \mathbf{v}_2) = \hat{p}(\mathbf{v}_1, \mathbf{v}_2)$;

3. Decode learned latents into observed variables: $[\hat{\mathbf{v}}_1, \hat{\mathbf{v}}_2] = \hat{g}([\hat{\mathbf{s}}_1, \hat{\mathbf{z}}, \hat{\mathbf{s}}_2])$, where $\hat{\mathbf{v}} \equiv \mathbf{v}$.

The key procedure that transforming ground truth latents to learned latents are the first two steps, denoted as $h = \hat{g}^{-1} \circ g$. To prove $\mathbf{s}_1 \sim \mathbf{s}'_1, \mathbf{z} \sim \mathbf{z}', \mathbf{s}_2 \sim \mathbf{s}'_2$, it is equivalent to prove that the Jaccobian matrix $J_h = \frac{\partial \hat{\mathbf{s}}}{\partial \mathbf{s}}$ has non-zero elements only between each corresponding pair. Other elements including $\frac{\partial \hat{\mathbf{s}}_2}{\partial \mathbf{s}_1}, \frac{\partial \hat{\mathbf{s}}_1}{\partial \mathbf{s}_2}, \frac{\partial \hat{\mathbf{s}}_1}{\partial \mathbf{z}}, \frac{\partial \hat{\mathbf{s}}_2}{\partial \mathbf{z}}, \frac{\partial \hat{\mathbf{z}}}{\partial \mathbf{s}_1}$ and $\frac{\partial \hat{\mathbf{z}}}{\partial \mathbf{s}_2}$ should always be zero. We split them into 3

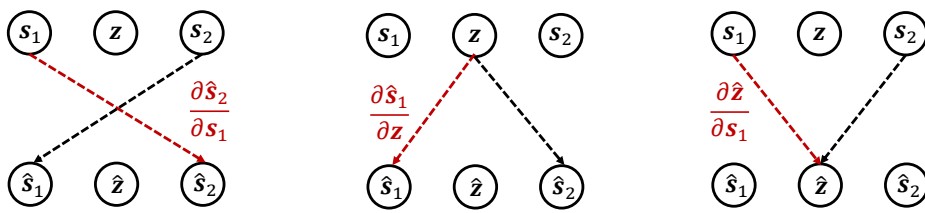

Figure 9: Non-existent edges to be proved.

groups and provide the proofs respectively (see Fig. 9 for illustration). We use the notation $\frac{\partial \mathbf{y}}{\partial \mathbf{x}} \equiv 0$ to represent that $\frac{\partial y[i]}{\partial x[j]}|_{x[j]=a} = 0$ for any $i$, $j$, and $a$.

**Step 1. prove that $\frac{\partial \hat{\mathbf{s}}_2}{\partial \mathbf{s}_1} \equiv 0$**

We first prove that $\frac{\partial \hat{\mathbf{s}}_2}{\partial \mathbf{s}_1} \equiv 0$, and $\frac{\partial \hat{\mathbf{s}}_1}{\partial \mathbf{s}_2} \equiv 0$ can be proved similarly. Assume there exist index $i$, $j$, and value $a$ such that $\frac{\partial \hat{s}_2[i]}{\partial s_1[j]}|_{s_1[j]=a} \neq 0$, consider the following operation: perturbing $s_1[j]$ by a small enough value $\epsilon$ while keeping other variables in $\mathbf{s}$ unchanged, then $\hat{\mathbf{s}}_2$ will change accordingly. In this case, if $\hat{\mathbf{s}}_1$ or $\hat{\mathbf{z}}$ also changes, this will violate assumption $ii$ (independence) according to Lemma A.1; if both $\hat{\mathbf{s}}_1$ and $\hat{\mathbf{z}}$ do not change, then $\hat{\mathbf{v}}_1$ do not change since both inputs keeps unchanged, and $\hat{\mathbf{v}}_2$ do not change since $\frac{\partial \mathbf{v}_2}{\partial \mathbf{s}_1} \equiv 0$ and $\hat{\mathbf{v}}_2 \equiv \mathbf{v}_2$, this means two different values of $\hat{\mathbf{s}}$ correspond to the same $\hat{\mathbf{v}}$, which violates assumption $i$ (invertibility). As a result, we have $\frac{\partial \hat{s}_2[i]}{\partial s_1[j]}|_{s_1[j]=a} \neq 0$ for any $i$, $j$, and $a$, thus $\frac{\partial \hat{\mathbf{s}}_2}{\partial \mathbf{s}_1} \equiv 0$.

**Step 2. prove that $\frac{\partial \hat{\mathbf{s}}_1}{\partial \mathbf{z}} \equiv 0$**

We then prove that $\frac{\partial \hat{\mathbf{s}}_1}{\partial \mathbf{z}} \equiv 0$, and $\frac{\partial \hat{\mathbf{s}}_2}{\partial \mathbf{z}} \equiv 0$ can be proved similarly. Assume there exist index $i$, $j$, and value $a$ such that $\frac{\partial \hat{s}_1[i]}{\partial z[j]}|_{z[j]=a} \neq 0$. At this time, denote $\hat{\mathbf{t}} = [\hat{\mathbf{z}}, \hat{\mathbf{s}}_2]$, then $\frac{\partial \hat{t}[k]}{\partial z[j]}|_{z[j]=b} = 0$ should hold for any $k$ and $b$ according to assumption $ii$ (independence) and Lemma A.1, i.e., $\frac{\partial \hat{\mathbf{t}}}{\partial z[j]} \equiv 0$. As a result, there is $\frac{\partial \hat{\mathbf{v}}_2}{\partial z[j]} = \frac{\partial \hat{\mathbf{v}}_2}{\partial \hat{\mathbf{t}}} \cdot \frac{\partial \hat{\mathbf{t}}}{\partial z[j]} = 0$, so that $\mathbf{v}_2 = \hat{\mathbf{v}}_2$ is not influenced by $z[j]$.

Therefore, we can remove dimension $j$ from $\mathbf{z}$ and merge $z[j]$ into $\mathbf{s}_1$, since $z[j]$ does not influence $\mathbf{v}_2$ and $z[j] \perp \mathbf{z}\backslash z[j]$. Now assumption $iii$ (minimality) of $\mathbf{z}$ is violated, as a result, we know $\frac{\partial \hat{s}_1[i]}{\partial z[j]}|_{z[j]=a} = 0$ is true for any $i$, $j$, and $a$, i.e., $\frac{\partial \hat{\mathbf{s}}_1}{\partial \mathbf{z}} \equiv 0$.

**Step 3. prove that $\frac{\partial \hat{\mathbf{z}}}{\partial \mathbf{s}_1} \equiv 0$**

At last, We prove that $\frac{\partial \hat{\mathbf{z}}}{\partial \mathbf{s}_1} \equiv 0$ and $\frac{\partial \hat{\mathbf{z}}}{\partial \mathbf{s}_2} \equiv 0$. Assume there exist index $i$, $j$, and value $a$ such that $\frac{\partial \hat{z}[j]}{\partial s_1[i]}|_{s_1[i]=a} \neq 0$, we can similarly get that, $\frac{\partial \hat{s}_1[k]}{\partial s_1[i]}|_{s_1[i]=b} = 0$ holds for any $k$ and $b$ according to assumption $ii$ (independence) and Lemma A.1, i.e., $\frac{\partial \hat{\mathbf{s}}_1}{\partial s_1[i]} \equiv 0$. Gathering all such dimensions in $\mathbf{s}_1$ and $\mathbf{s}_2$, and merge them into one variable $\mathbf{s}_0$. In this case, $\mathbf{s}_0$ and $\mathbf{z}$ are $d$-separated from $\hat{\mathbf{s}}_1$ and $\hat{\mathbf{s}}_2$ by $\hat{\mathbf{z}}$ on the graph of $J_h$ (see Fig. 10), so that $[\mathbf{s}_0, \mathbf{z}] \sim \hat{\mathbf{z}}$ according to assumption $i$ (invertibility), thus $\mathbf{z} \prec \hat{\mathbf{z}}$, i.e., assumption $iii$ (minimality) of $\hat{\mathbf{z}}$ is violated. As a result, we have $\frac{\partial \hat{\mathbf{z}}}{\partial \mathbf{s}_1} \equiv 0$ and $\frac{\partial \hat{\mathbf{z}}}{\partial \mathbf{s}_2} \equiv 0$. $\square$

### A.4 PROOF OF PROPOSITION. 5.1

**Proof.** Since $\mathbf{z}$ is not minimal, we can find a model $\hat{M}$ satisfying assumption $i$ and $ii$, such that $\hat{p}(\hat{\mathbf{v}}_1, \hat{\mathbf{v}}_2) = p(\mathbf{v}_1, \mathbf{v}_2)$ for any $(\hat{\mathbf{v}}_1, \hat{\mathbf{v}}_2) = (\mathbf{v}_1, \mathbf{v}_2) \in \mathcal{V}$ and $\hat{\mathbf{z}} \prec \mathbf{z}$. Since both $M$ and $\hat{M}$ satisfy the invertibility assumption and shares the same marginal distribution over the observable variables, we can build a similar data generation process as in Fig. 8 (To ensure that each dimensions of $\mathbf{s}$ are independent, nolinear ICA is also applied to $\mathbf{s}_1$, $\mathbf{z}$, and $\mathbf{s}_2$, respectively, and then constant dimensions are also removed, which does not influence the conclusion). Then according to the proofs of Thm. 1

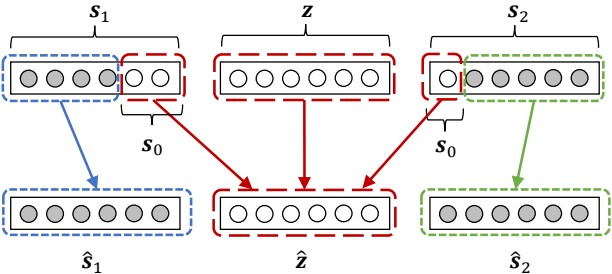

Figure 10: Graph of $J_h$, on which $\mathbf{s}_0$ and $\mathbf{z}$ are $d$-separated from $\hat{\mathbf{s}}_1$ and $\hat{\mathbf{s}}_2$ by $\hat{\mathbf{z}}$.

(step 1.), there is $\frac{\partial \hat{\mathbf{s}}_2}{\partial \mathbf{s}_1} \equiv 0$ and $\frac{\partial \hat{\mathbf{s}}_1}{\partial \mathbf{s}_2} \equiv 0$. According to whether $\frac{\partial \hat{\mathbf{s}}_1}{\partial \mathbf{z}} \equiv 0$ and $\frac{\partial \hat{\mathbf{s}}_2}{\partial \mathbf{z}} \equiv 0$ hold (step 2.), we discuss the following two cases.

Case 1: $\frac{\partial \hat{\mathbf{s}}_1}{\partial \mathbf{z}} \equiv 0$ and $\frac{\partial \hat{\mathbf{s}}_2}{\partial \mathbf{z}} \equiv 0$. In this case, we then consider step 3 in proofs of Thm. 1. If $\frac{\partial \hat{\mathbf{z}}}{\partial \mathbf{s}_1} \equiv 0$ and $\frac{\partial \hat{\mathbf{z}}}{\partial \mathbf{s}_2} \equiv 0$, then all extra edges between $\mathbf{z}$ and $\hat{\mathbf{z}}$ are excluded, which means $\hat{\mathbf{z}} \sim \mathbf{z}$, causing contradiction to known condition $\hat{\mathbf{z}} \prec \mathbf{z}$; If not, their would be $\mathbf{z} \prec \hat{\mathbf{z}}$ according to the conclusion from step 3, which also causes contradiction. As a result, this case does not exist.

Case 2: $\frac{\partial \hat{\mathbf{s}}_1}{\partial \mathbf{z}} \not\equiv 0$ or $\frac{\partial \hat{\mathbf{s}}_2}{\partial \mathbf{z}} \not\equiv 0$. We assume $\frac{\partial \hat{s}_1[i]}{\partial z[j]}\big|_{z[j]=a} \neq 0$ w.l.o.g.. In this case, we can remove dimension $j$ from $\mathbf{z}$ and merge $z[j]$ into $\mathbf{s}_1$ to get a new model without changing the observable distribution according to the proofs of Thm. 1 (step 2.). As a result, merge all dimensions $j$ in $\mathbf{z}$ such that $\frac{\partial \hat{s}_1[i]}{\partial z[j]} \not\equiv 0$ (for any $i$) to get a new variable $\mathbf{z}_1^{(1)}$, and merge all dimensions $k$ in $\mathbf{z}$ such that $\frac{\partial \hat{s}_2[i]}{\partial z[k]} \not\equiv 0$ (for any $i$) to get a new variable $\mathbf{z}_2^{(1)}$, the rest dimensions in $\mathbf{z}$ constitute $\mathbf{z}_0^{(1)}$. If any of the above cases does not exist, set the corresponding variable as constant. Note that $\mathbf{z}_1$ and $\mathbf{z}_2$ cannot be constant at the same time. Now we get an new model $\hat{\mathcal{M}}^{(1)}$ with $\hat{\mathbf{s}}_1^{(1)} = [\mathbf{s}_1, \mathbf{z}_1^{(1)}]$, $\hat{\mathbf{z}}^{(1)} = \mathbf{z}_0^{(1)}$, $\hat{\mathbf{s}}_2^{(1)} = [\mathbf{s}_2, \mathbf{z}_2^{(1)}]$, and $\hat{\mathcal{M}}^{(1)}$ shares the same observational distribution with $\hat{\mathcal{M}}$ and $\mathcal{M}$. Since $\mathbf{z}$ has non-constant independent dimensions, so that $\mathrm{IDim}(\mathbf{z}_0^{(1)}) = \dim(\mathbf{z}_0^{(1)}) < \dim(\mathbf{z}) = \mathrm{IDim}(\mathbf{z})$.

Now check if model $\hat{\mathcal{M}}^{(1)}$ satisfy the minimality condition. If not, we replace $\hat{\mathcal{M}}$ with $\hat{\mathcal{M}}^{(1)}$, then repeat all above procedures from the very beginning, and get model $\hat{\mathcal{M}}^{(2)}$. Repeat this process till $\hat{\mathcal{M}}^{(K)}$ satisfy the minimality condition. $K$ is finite since the dimension of $\hat{\mathbf{z}}^{(k)}$ decreases for at least 1 during each repetition, and finally a constant $\hat{\mathbf{z}}^{(k)}$ undoubtedly satisfy the minimality condition. Now we get the target $\mathcal{M}' = \hat{\mathcal{M}}^{(K)}$ with $\mathbf{s}_1' \sim [\mathbf{s}_1, \mathbf{z}_1]$, $\mathbf{z}' \sim \mathbf{z}_0$, $\mathbf{s}_2' \sim [\mathbf{s}_2, \mathbf{z}_2]$, where $\mathbf{z}_0 = \mathbf{z}_0^{(K)}$, $\mathbf{z}_1 = [\{\mathbf{z}_1^{(k)}\}_{k=1}^K]$, $\mathbf{z}_2 = [\{\mathbf{z}_2^{(k)}\}_{k=1}^K]$, and $\mathrm{IDim}(\mathbf{z}_0) < \mathrm{IDim}(\mathbf{z})$. □

## A.5 Proofs of Corollary 5.1

**Proof.** We only need to prove that model $\hat{\mathcal{M}}$ satisfy the minimality assumption. Assume $\hat{\mathcal{M}}$ does not satisfy the minimality assumption, then according to Proposition 5.1, we can get an identifiable model $\mathcal{M}'$ satisfying Assumption 1 by reducing the intrinsic dimension of $\hat{\mathbf{z}}$ without changing the marginal distribution of observable variables, i.e. $\mathrm{IDim}(\mathbf{z}') < \mathrm{IDim}(\hat{\mathbf{z}}) \leq \dim(\hat{\mathbf{z}})$. According to Thm. 1, there is $\mathbf{z}' \sim \mathbf{z}$, so that $\mathrm{IDim}(\mathbf{z}) = \mathrm{IDim}(\mathbf{z}') < \dim(\hat{\mathbf{z}})$, which violates the known condition $\dim(\hat{\mathbf{z}}) = \mathrm{IDim}(\mathbf{z})$. As a result, we know model $\hat{\mathcal{M}}$ satisfy the minimality assumption. Then by applying Thm. 1 again, we get $\mathbf{s}_1 \sim \hat{\mathbf{s}}_1$, $\mathbf{z} \sim \hat{\mathbf{z}}$, $\mathbf{s}_2 \sim \hat{\mathbf{s}}_2$. □

## A.6 Lemmas for Thm 2

**Lemma A.2** *(Intermediate variables identifiability) For a general model $(p_s, \{g_j\}_{j=1}^n)$ satisfying Assumption 2, a basis model $(\hat{p}_{s_1}, \hat{p}_z, \hat{p}_{s_2}, \hat{g}_1, \hat{g}_2)$, and a given index $j \in \{1, \cdots, n\}$, if $\hat{p}(\hat{\mathbf{v}}_1 = \mathbf{v}_j, \hat{\mathbf{v}}_2 = \mathbf{v}\backslash\{\mathbf{v}_j\}) = p(\mathbf{v})$ for any $\mathbf{v} \in \mathcal{V}$, then there is $\hat{\mathbf{s}}_1 \sim \mathbf{s}_{2^j}$, $\hat{\mathbf{z}} \sim [\mathbf{s}_i | i \neq 2^j, i\&2^j \neq 0]$, $\hat{\mathbf{s}}_2 \sim [\mathbf{s}_i | i\&2^j = 0]$.*

**Proof.** Consider a basis model $\mathcal{M}' = (p'_{s_1}, p'_z, p'_{s_2}, g'_1, g'_2)$ with

$$\mathbf{s}'_1 = \mathbf{s}_{2^j},$$
$$\mathbf{z}' = [\mathbf{s}_i | i \neq 2^j, i \& 2^j \neq 0],$$
$$\mathbf{s}'_2 = [\mathbf{s}_i | i \& 2^j = 0],$$
$$\mathbf{v}'_1 = g'_1(\mathbf{s}'_1, \mathbf{z}') = g_1(\{\mathbf{s}_i | i \& 2^j \neq 0\}) = \mathbf{v}_j,$$
$$\mathbf{v}'_2 = g'_2(\mathbf{z}', \mathbf{s}'_1) = [g_k(\{\mathbf{s}_i | i \& 2^k \neq 0\}) | k \neq j] = [\mathbf{v}_k | k \neq j],$$

this model shares the same marginal distribution $p(\mathbf{v}_1, \mathbf{v}_2)$ with model $\hat{\mathcal{M}} = (\hat{p}_{s_1}, \hat{p}_z, \hat{p}_{s_2}, \hat{g}_1, \hat{g}_2)$. To prove this proposition, we only need to prove that model $\mathcal{M}'$ is identifiable, i.e. satisfying Assumption 1.

The invertibility inherits from general model $\mathcal{M}$, since $[\mathbf{s}'_1, \mathbf{z}', \mathbf{s}'_2] \sim [\{\mathbf{s}_i\}_{i=1}^m]$ and $[\mathbf{v}'_1, \mathbf{v}'_2] \sim [\{\mathbf{v}_j\}_{j=1}^n]$. The independence holds since all latent variables in $\mathcal{M}'$ are concatenations of mutually independent variables, and $\mathbf{s}_{2^j} \notin \{\mathbf{s}_i | i \neq 2^j, i \& 2^j \neq 0\}$, $\mathbf{s}_{2^j} \notin \{\mathbf{s}_i | i \& 2^j = 0\}$, $\{\mathbf{s}_i | i \neq 2^j, i \& 2^j \neq 0\} \cap \{\mathbf{s}_i | i \& 2^j = 0\} = \emptyset$. We then focus on the minimality property.

Assume $\mathcal{M}'$ does not satisfy the minimality condition in Assumption 1, then according to Proposition 5.1, some dimensions in $\mathbf{z}'$ can be moved into $\mathbf{s}'_1$ or $\mathbf{s}'_2$ to get an identifiable model. Since $\mathbf{z}'$ is a concatenation of some $\mathbf{s}_i$, we track the moved dimensions are from which $\mathbf{s}_i$, and denote the set of such $\mathbf{s}_i$ as $S$. Now we only need to prove that there exists $\mathbf{s}_l \in S$ which is not minimal and violates the hierarchical minimality condition in Assumption 2.

For better understanding, we call variable $\mathbf{s}_k$ an "upper" variable w.r.t. $\mathbf{s}_l$, iff. $k \neq l$ and $k \& l = l$, i.e. in binary representation, $k$ has value 1 wherever $l$ has value 1. Recall how to violate the hierarchical minimality condition in Assumption 2, that is, $\mathbf{s}_l$ has more than 2 descendants, and is not minimal while all its upper variables are minimal. According to the numbering rule, all variables in $\{\mathbf{s}_i | i \neq 2^j, i \& 2^j \neq 0\}$ have indices with two or more "1"s in binary representation, and thus have more than 2 descendants. Pick a locally "upperest" variable in $S$ as the chosen $\mathbf{s}_l$, i.e., there does not exist another $\mathbf{s}_k \in S$ which is an upper variable w.r.t. $\mathbf{s}_l$. This operation is possible since the "upper" relation is a partial order define on a finite set. Now we only need to prove that, no moved dimensions from $\mathbf{s}_l$ will go to its upper variable, or equivalently, $\mathbf{s}_{2^j}$ and elements in $\{\mathbf{s}_i | i \& 2^j = 0\}$ are not upper variables w.r.t. $\mathbf{s}_l$.

For $\mathbf{s}_{2^j}$, since $l \& 2^j \neq 0$ which means $l \& 2^j = 2^j$, i.e., $\mathbf{s}_l$ is an upper variable w.r.t. $\mathbf{s}_{2^j}$, so that $\mathbf{s}_{2^j}$ is not an upper variable w.r.t. $\mathbf{s}_l$. For elements in $\{\mathbf{s}_i | i \& 2^j = 0\}$ such as $\mathbf{s}_i$, since $i$ takes value 0 on $j$-th digit in binary representation but $l$ takes value 1, then $i \& l$ cannot take value 1 on position $j$, thus $i \& l \neq l$, i.e. $\mathbf{s}_i$ is not an upper variable.

Gathering all results up, we know no moved dimensions from $\mathbf{s}_l$ will go to its upper variable, so that $\mathbf{s}_l$ violates the hierarchical minimality assumption, which means the initial assumption is not true, i.e., $\mathcal{M}'$ satisfies the minimality condition in Assumption 1. As a result, $\mathcal{M}'$ is identifiable and thus latent variables in model $\hat{\mathcal{M}}$ are equivalent to that in $\mathcal{M}'$, i.e.,

$$\hat{\mathbf{s}}_1 \sim \mathbf{s}'_1 = \mathbf{s}_{2^j},$$
$$\hat{\mathbf{z}} \sim \mathbf{z}' = [\mathbf{s}_i | i \neq 2^j, i \& 2^j \neq 0],$$
$$\hat{\mathbf{s}}_2 \sim \mathbf{s}'_2 = [\mathbf{s}_i | i \& 2^j = 0].$$

$\square$

**Definition A.1** (*Intersection of random variables*). *For two random variables $\mathbf{x}$ and $\mathbf{y}$ with joint distribution $p(\mathbf{x}, \mathbf{y})$, we say the intersection of $\mathbf{x}$ and $\mathbf{y}$ exists iff. there exists a basis model $(p_{s_1}, p_z, p_{s_2}, g_1, g_2)$ satisfying Assumption 1, such that $p(\mathbf{v}_1 = \mathbf{x}, \mathbf{v}_2 = \mathbf{y}) \equiv p(\mathbf{x}, \mathbf{y})$, and all solutions of the intersection constitute the equivalent set w.r.t. $\mathbf{z}$, denoted as $\mathbf{x} \sqcap \mathbf{y} \sim \mathbf{z}$.*

**Discussion.** With our identifiability theory, we are able to define the "intersection" operation for two random variables, whose result is a variable representing their share information, i.e. an equivalent class w.r.t. $\mathbf{z}$. Our assumptions provide a sufficient condition for the validity of such intersection operation.

**Lemma A.3** *(Concatenated latents identifiability) Given a set of mutually independent variables $S = \{\mathbf{s}_i\}_{i=1}^m$, for any subset $A, B \in \mathcal{P}(S)$, the intersection of $[A]$ and $[B]$ exists, there is $[A] \sqcap [B] \sim [A \cap B]$.*

**Proof.** Consider the following basis model $\mathcal{M}$:

$$\begin{aligned}
\mathbf{s}_1 &= [A \backslash B], \\
\mathbf{z} &= [A \cap B], \\
\mathbf{s}_2 &= [B \backslash A], \\
\mathbf{v}_1 &= [\mathbf{s}_1, \mathbf{z}] = [A], \\
\mathbf{v}_2 &= [\mathbf{z}, \mathbf{s}_2] = [B].
\end{aligned}$$

The marginal distribution of $\mathbf{v}_1, \mathbf{v}_2$ for model $\mathcal{M}$ is exactly $p([A], [B])$, so that we only need to prove that $\mathcal{M}$ is identifiable, i.e. satisfying Assumption 1.

The invertibility condition is satisfied, since $[\mathbf{s}_1, \mathbf{z}, \mathbf{s}_2] \sim [A \cup B] = [\mathbf{v}_1, \mathbf{v}_2]$. The independence condition is satisfied, since all elements in $A \cup B$ are mutually independent and $(A \backslash B) \cap (A \cap B) = \emptyset, (A \cap B) \cap (B \backslash A) = \emptyset, (A \backslash B) \cap (B \backslash A) = \emptyset$.

Assume the minimality condition does not hold, then there exist $[\mathbf{z}_0, \mathbf{z}_1, \mathbf{z}_2] \sim \mathbf{z}$, $\mathbf{s}_1' \sim [\mathbf{s}_1, \mathbf{z}_1]$, $\mathbf{z}' \sim \mathbf{z}_0$, $\mathbf{s}_2' \sim [\mathbf{s}_2, \mathbf{z}_2]$ according to Proposition 5.1. $\mathbf{z}_1$ and $\mathbf{z}_2$ cannot be constant simultaneously, we assume $\mathbf{z}_1$ is not constant w.l.o.g., then $\mathbf{v}_2 \not\perp \mathbf{z}_1$ since $\mathbf{z}_1 \prec \mathbf{z} \prec \mathbf{v}_2$. However, $\mathbf{s}_1'$ also contains information about $\mathbf{z}_1$, so that $\mathbf{v}_2 \not\perp \mathbf{s}_1'$, which violates the Markov assumption of a causal model. As a result, we know $\mathcal{M}$ satisfies the minimality condition.

Finally, we know $\mathcal{M}$ is identifiable, so that $[A] \sqcap [B] = \mathbf{v}_1 \sqcap \mathbf{v}_2 \sim \mathbf{z} = [A \cap B]$. $\qquad\square$

A.7   PROOFS OF THM. 2

**Proof.** We give a constructive proof here, such that a general PBG-SCM can be identified by iteratively applying the basis model identification algorithm. Given the observable variables $\mathbf{v} = (\mathbf{v}_1, \cdots, \mathbf{v}_n)$ under an identifiable model $(p_s, \{g_j\}_{j=1}^n)$ satisfying Assumption 2, the whole process include 2 steps (Alg. 1):

**Step 1. Identify the intermediate variables.**

Consider setting $\mathbf{v}_j$ as one observed variable and concatenation of the rest $\mathbf{v} \backslash \{\mathbf{v}_j\} = [\cdots, \mathbf{v}_{j-1}, \mathbf{v}_{j+1}, \cdots]$ as another observed variable. Apply the identification algorithm for basis model, we can get the identified latents $\hat{\mathbf{s}}_1 \sim \mathbf{s}_{2^j}$, $\hat{\mathbf{z}} \sim [\mathbf{s}_i | i \neq 2^j, i \& 2^j \neq 0]$, $\hat{\mathbf{s}}_2 \sim [\mathbf{s}_i | i \& 2^j = 0]$ according to Lemma A.2. We define the intermediate variables $\mathbf{t}_j^+$ and $\mathbf{t}_j^-$ as the following:

$$\begin{aligned}
\mathbf{t}_j^+ &= [\hat{\mathbf{s}}_1, \hat{\mathbf{z}}] \sim [\mathbf{s}_i | i \& 2^j \neq 0], \\
\mathbf{t}_j^- &= \hat{\mathbf{s}}_2 \sim [\mathbf{s}_i | i \& 2^j = 0].
\end{aligned} \tag{3}$$

$\mathbf{t}_j^+$ is equivalent to the concatenation of all $\mathbf{s}_i$ which is connected with $\mathbf{v}_j$ in the causal graph, while the corresponding $\mathbf{t}_j^-$ is equivalent to the concatenation of all $\mathbf{s}_i$ which have no connection with $\mathbf{v}_j$.

**Step 2. Identify the latent variables.**

Consider applying the basis model by setting $\mathbf{v}_1 = \mathbf{t}_i^+$ and $\mathbf{v}_2 = \mathbf{t}_j^-, j \neq i$, we can run the identification algorithm to get $\mathbf{v}_1 \sqcap \mathbf{v}_2 \sim [\mathbf{s}_k | k \& 2^i \neq 0, k \& 2^j = 0]$ according to Lemma A.3. Similarly, to identify the latent variable $\mathbf{s}_i$, we need to do intersection over all $\mathbf{t}_j$ according to the binary representation of index $i$: if $j \& 2^i \neq 0$, take intersection over $\mathbf{t}_j^+$, otherwise over $\mathbf{t}_j^-$. For example, in a PBG-SCM with 3 observable variables, $\mathbf{s}_1 \sim \mathbf{t}_1^+ \sqcap \mathbf{t}_2^- \sqcap \mathbf{t}_3^-$ since the binary representation of 1 under length 3 is $[001]_2$.

**Discussion.** Note that Alg. 1 mainly serves as part of the constructive proof for Thm. 2 and can be further improved. Its time complexity is $O(m \log m)$, and can be optimized to $O(m)$ by storing the intermediate results, with the cost of increasing space complexity from $O(\log m)$ to $O(m)$. Even though this algorithm may still be time-consuming in practice since $m = 2^n - 1$, and we suggest to seek alternative solutions.

---

**Algorithm 1** Identification Algorithm for General PBG-SCM

---

**Input:** Basis model identification algorithm BM, observable variables $\mathbf{v} = (\mathbf{v}_1, \cdots, \mathbf{v}_n)$
**Output:** Latent variables $\mathbf{s} = (\mathbf{s}_1, \cdots, \mathbf{s}_m)$
 1: **for** $j = 1$ to $n$ **do**
 2:   $\hat{\mathbf{s}}_1, \hat{\mathbf{z}}, \hat{\mathbf{s}}_2 \leftarrow \text{BM}(\mathbf{v}_j, \mathbf{v}\backslash\{\mathbf{v}_j\})$
 3:   $\mathbf{t}_j^+ \leftarrow [\hat{\mathbf{s}}_1, \hat{\mathbf{z}}], \mathbf{t}_j^- \leftarrow \hat{\mathbf{s}}_2$
 4: **end for**
 5: **for** $i = 1$ to $m$ **do**
 6:   **if** $i\%2 = 0$ **then** $\mathbf{s}_i \leftarrow \mathbf{t}_1^-$ **else** $\mathbf{s}_i \leftarrow \mathbf{t}_1^+$ **end if**
 7:   **for** $i = 2$ to $n$ **do**
 8:    **if** $i\&2^j = 0$ **then** $\hat{\mathbf{s}}_1, \hat{\mathbf{z}}, \hat{\mathbf{s}}_2 \leftarrow \text{BM}(\mathbf{s}_i, \mathbf{t}_j^-)$ **else** $\hat{\mathbf{s}}_1, \hat{\mathbf{z}}, \hat{\mathbf{s}}_2 \leftarrow \text{BM}(\mathbf{s}_i, \mathbf{t}_j^+)$ **end if**
 9:    $\mathbf{s}_i \leftarrow \hat{\mathbf{z}}$
10:   **end for**
11: **end for**
12: **return** $\mathbf{s}_1, \cdots, \mathbf{s}_m$

---

### A.8 DETAILED EXPERIMENT SETTINGS

**Definition of $R^2$ scores.** The original $R^2$ score is for evaluation of the degree of relation "$\mathbf{y} \preceq \mathbf{x}$". Given a set of observations $\{(\mathbf{x}^{(i)}, \mathbf{y}^{(i)})\}_{i=1}^N$ where $\mathbf{x}^{(i)} = (x_1^{(i)}, \cdots, x_K^{(i)})$, $\mathbf{y}^{(i)} = (y_1^{(i)}, \cdots, y_L^{(i)})$, $R^2$ first constructs an optimal prediction model $\hat{\mathbf{y}} = g_\theta(\mathbf{x})$ in terms of mean squared error $\text{MSE} = \mathbb{E}_{\mathbf{x},\mathbf{y}} \|g_\theta(\mathbf{x}) - \mathbf{y}\|_2^2$. Then the $R^2$ score is

$$R_{x \to y}^2 = 1 - \frac{1}{N} \sum_{i=1}^N \frac{\sum_{j=1}^L (\hat{y}_j^{(i)} - y_j^{(i)})^2}{\sum_{j=1}^L (y_j^{(i)} - \bar{y}_j)^2},$$

where $\bar{y}_j$ is the mean of $y_j$. Intuitively, $R^2$ measures the difference in prediction loss between $\mathbf{x} \to \mathbf{y}$ and $\emptyset \to \mathbf{y}$. If $\mathbf{y} \preceq \mathbf{x}$, then $R^2$ reaches its maximal value 1; otherwise $R^2 \in [0, 1)$.

For evaluation of variable equivalence such as $\hat{\mathbf{z}} \sim \mathbf{z}$, or variable independence such as $\mathbf{s}_1 \perp \mathbf{s}_2$, we use F1 score-like expressions to emphasize the worse side:

$$R_{\hat{\mathbf{z}} \sim \mathbf{z}}^2 = \frac{2 \cdot R_{\hat{\mathbf{z}} \to \mathbf{z}}^2 \cdot R_{\mathbf{z} \to \hat{\mathbf{z}}}^2}{R_{\hat{\mathbf{z}} \to \mathbf{z}}^2 + R_{\mathbf{z} \to \hat{\mathbf{z}}}^2}, \quad R_{\mathbf{s}_1 \perp \mathbf{s}_2}^2 = \frac{2 \cdot (1 - R_{\mathbf{s}_1 \to \mathbf{s}_2}^2) \cdot (1 - R_{\mathbf{s}_2 \to \mathbf{s}_1}^2)}{2 - R_{\mathbf{s}_1 \to \mathbf{s}_2}^2 - R_{\mathbf{s}_2 \to \mathbf{s}_1}^2}.$$

All such $R^2$ scores take best value at 1. For the prediction model $g_\theta$, we also implement it by MLP, and train the parameters $\theta$ till convergence.

**Configuration for MLP.** For the generation of synthetic data, we use MLPs with 2 layers, hidden size is set to 64, and activation function is *Tanh*. For the prediction model used in $R^2$ score, we use MLPs with 3 layers, hidden size is set to 1024, and activation function is *ReLU*. For each component in identification model, we use MLPs with 3 layers, hidden size is set to 512, and activation function is *ReLU*.

**Details of $R^2$ score calculation.** We use a total amount of 20,000 examples for evaluation, from which we sample 5,000 examples as test data, and the rest 15,000 examples as training data for the prediction model. Each variable is normalized to have 0 mean and unit standard deviation before evaluation. For the training of prediction model, we use AdamW optimizer with learning rate 1e-3, and applied an scheduler for decayed learning rate. We use 20% data as validation set and train the model for 500 epochs with early-stop strategy.

**Configuration and training for the identification model.** For the encoder, we concatenate all input variables and apply an MLP as encoder to get the concatenation of learned latent variables $[\mathbf{s}_1, \mathbf{z}, \mathbf{s}_2]$. Except for $d_z$ which has been specially discussed, we set $d_{s_1} = 3$ and $d_{s_2} = 4$, which is the same as ground truth model. Then we use two MLPs as decoders to get $\mathbf{v}_1$ and $\mathbf{v}_2$ as in Eqn. 1 or Eqn. 2. we follow literature (Cheng et al., 2020) to implement the CLUB model. We use a total amount of 100,000 examples for training identification model. We use AdamW optimizer with learning rate 1e-3. The weight of reconstruction loss is set to 10.0 and weight of independence loss (CLUB loss) is set to 0.01.

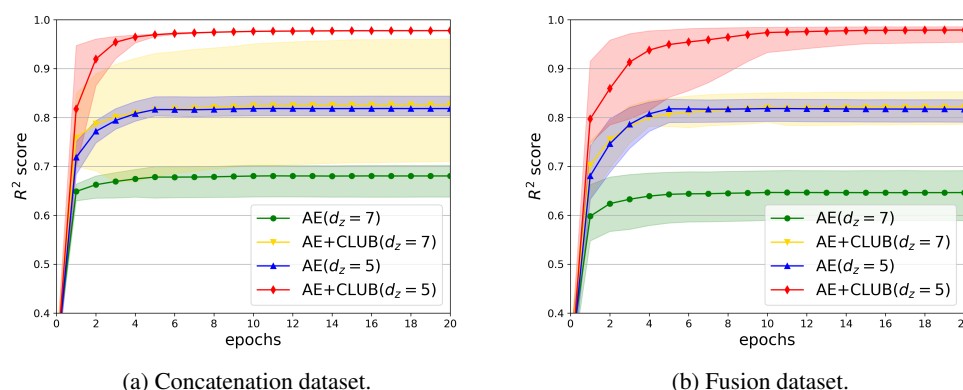

(a) Concatenation dataset.

(b) Fusion dataset.

Figure 11: The mean of averaged $R^2$ score curve during training of identification algorithms. Shaded area represents the region from min value to max value over 5 runs.

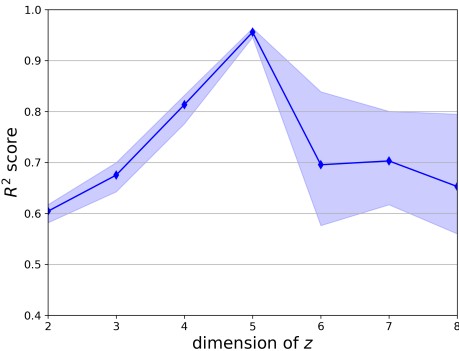

Figure 12: The influence of $d_z$ in the identification of basis model on the *Split* dataset. Shaded area represents the region from min value to max value over 5 runs.

**Hardware environment.** Our experiments are done with 1 NVIDIA Tesla K80 GPU.

### A.9 ADDITIONAL EXPERIMENT RESULTS

We show the training curve on *Concatenation* and *Fusion* dataset in Fig. 11. Similar conclusions can be derived as Fig. 3.

We further run an experiment to show the effect of $d_z$ in learning of basis models. We also use the *Split* dataset as well as the AE+CLUB method, and vary $d_z$ from 2 to 8. The identification result for latent variables is shown in Fig. 12. We can see that, when $d_z$ is the same as ground truth model, we can get the best identification result. If $d_z$ goes lower, then $R^2$ score decreases since the model have not enough capacity to achieve both invertibility and independence. If $d_z$ goes higher, then $R^2$ score also decreases, and the variance gets significantly larger, since $\mathbf{z}$ may contain uncertainly more information in this case.

To better illustrate the quality of disentanglement, we run an experiment on a synthetic image dataset, which we name as *TwoShapes*. This dataset is for validation of our basis model. $\mathbf{v}_1$ and $\mathbf{v}_2$ are both $32 \times 32$ grey scale images, while $\mathbf{v}_1$ contains a circle and $\mathbf{v}_2$ contains a triangle. For the latent variables, $\mathbf{s}_1$ is 2-dimentional which controls the foreground and background grey scale value of $\mathbf{v}_1$; $\mathbf{s}_2$ is 2-dimentional which controls the foreground and background grey scale value of $\mathbf{v}_2$; $\mathbf{z}$ is 2-dimentional which controls the vertical and horizontal coordinate of the shapes in both $\mathbf{v}_1$ and $\mathbf{v}_2$. See Fig. 13 for examples of this dataset.

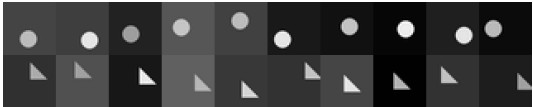

Figure 13: Examples from our synthetic *TwoShapes* dataset. Each column is one observation, the upper square represents $\mathbf{v}_1$, and the lower square represents $\mathbf{v}_2$.

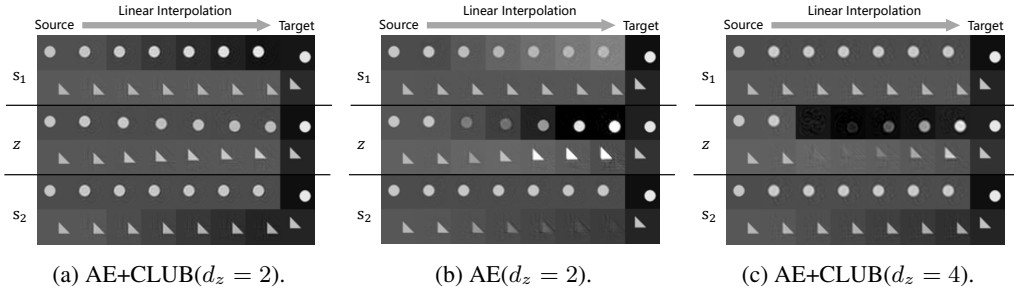

(a) AE+CLUB($d_z = 2$).  (b) AE($d_z = 2$).  (c) AE+CLUB($d_z = 4$).

Figure 14: Reconstructed images from interpolated latent representations between source and target images w.r.t. each latent variable. For each learned model, we show the interpolation results over $\mathbf{s}_1/\mathbf{z}/\mathbf{s}_2$ in the upper/middle/lower area, respectively.

We flatten each image into a 1-dimentional vector as observed variables, and then apply the same identification algorithms as in Sec. 6. After obtaining the latent variables, we select 2 images as source and target images, do linear interpolation over their $\mathbf{s}_1$, $\mathbf{z}$, $\mathbf{s}_2$ respectively (while keeping other latent variables unchanged), and show the reconstructed images $\mathbf{v}_1$ and $\mathbf{v}_2$. Results are shown in Fig. 14.

We can see that AE+CLUB($d_z = 2$) achieves an ideal disentanglement in Fig. 14a: $\mathbf{s}_1$ controls the foreground and background color of the upper image, $\mathbf{s}_2$ controls the foreground and background color of the lower image, $\mathbf{z}$ controls nothing more than the coordinates of the shapes in both images. These results support our conclusion that the model would be identifiable if invertibility/independence/minimality are all achieved. For comparison, other two methods achieve clearly inferior disentanglement. AE($d_z = 2$) is a method without independence constraint, in its results in Fig. 14b, $\mathbf{s}_1$ incorrectly controls the background color of upper image, $\mathbf{z}$ controls too many attributes besides the coordinates, and $\mathbf{s}_2$ incorrectly controls the foreground color of lower image. AE+CLUB($d_z = 4$) is a method that cannot guarantee the minimality constraint, in its results in Fig. 14c, $\mathbf{s}_1$ learns nothing, and $\mathbf{z}$ controls almost everything except the background color of lower image, which is luckily controlled by $\mathbf{s}_2$. These results further confirm our conclusion that besides invertibility, independence and minimality constraints are also necessary for successful identification.

