# OpenReview forum: "Latent Variable Identifiability in Nonlinear Causal Models with Single-domain Data under Minimality Condition"
_ICLR.cc/2025/Conference — Submitted to ICLR 2025_

### Official Review · Reviewer_iYjM · 2024-10-22

**Soundness:** 3
**Presentation:** 2
**Contribution:** 1
**Rating:** 3
**Confidence:** 4

**Summary:**

This paper provides an identification result of latent variables in nonlinear causal models with single-domain data.

**Strengths:**

1. The theoretical results presented in this paper are sound, the authors provide detailed proofs.

2. This paper is well-organized and easy to follow.

**Weaknesses:**

1. The result of this paper is too weak, making it insignificant. To simplify the identification problem, it assumes that there is no direct path between observed variables. Even so, it can only identify latent variables in the PBG-SCM obtained by reducing the original SCM substantially. Notably, a PBG-SCM usually corresponds to a large number of SCMs. A latent variable in a PBG-SCM corresponds to the combination of several exogenous noises in the original SCM, which lacks physical meaning. In fact, even the number of latent variables in an SCM is not equal to that in the corresponding PBG-SCM. Therefore, identification of latent variables in the PBG-SCM is far from identification of latent variables in the original SCM, let alone causal relations between latent variables. I list two previous works for comparison. [1] also can only identify a simplified SCM rather than the original SCM, but it does not make the assumption that there is no direct path between observed variables. Moreover, the number of latent variables and causal relations between observed variables in the simplified SCM are same as that in the original SCM. [2] also makes the assumption that there is no direct path between observed variables, but it can identify latent variables and their causal relations.

2. In line 77, the author claims that "all these assumptions are mild, and necessary". First, the authors do not verify that all assumptions are mild, for instance, they are satisfied in most real-world cases or they are the same as or milder than assumptions made by related works. Specifically, I agree that the invertibility assumption is mild because it is used by most related works, but I cannot agree that the minimality assumptions are mild. Second, the authors only prove that minimality assumption is necessary in Proposition 5.1, but the other two assumptions are not proven to be necessary.

3. It seems that the identification algorithm based on auto encoder needs to know the dimension of $z$, but in real-world scenarios, this is usually unknown. Given an inaccurate dimension of $z$, the performance is not satisfactory.

4. Some minor concern: It world be better to give a proof sketch for Theorem 1 and provide some experimental results on real-world dataset.

[1] Hoyer P O, Shimizu S, Kerminen A J, et al. Estimation of causal effects using linear non-Gaussian causal models with hidden variables. International Journal of Approximate Reasoning, 2008, 49(2): 362-378.

[2] Kong L, Huang B, Xie F, et al. Identification of nonlinear latent hierarchical models. Advances in Neural Information Processing Systems, 2023, 36: 2010-2032.

**Questions:**

According to [1], SCM consists of a series of assignments and a joint probability of exogenous noises, but in this paper, SCM is a triplet (G,F,P), the authors should list their references.

[1] Peters J, Janzing D, Schölkopf B. Elements of causal inference: foundations and learning algorithms. The MIT Press, 2017.

---

> ### Author Response · Authors · 2024-11-27
>
> We first clarify a potential misunderstanding here, before going into the details. This paper focuses on identification of latent variables, rather than causal structures, and there are abundant works focusing on the same topic (See “Latent variable identifiability” section in our Related Works). Also, we did not claim any new identification algorithm as our contribution, neither did most related works on latent variable identifiability theory (not those on structure identification).
>
> **For weakness 1:**
>
> Identification of the latent variables in an SCM is never an easy work. It is a consensus that there is no free lunch: to make a model identifiable, assumptions are necessary, and stronger results require stricter assumptions. The assumption “there is no direct path between observed variables” has been adopted by many works (see line 155-157 for related works), and is reasonable and necessary in our setting (see our reply to reviewer aNXT for an explanation, question 5). We do not aim to identify *all* latent variables in the original SCM, and this is not one of our main claims. Instead, the reduction process explains why we introduce the PBG structure, and builds a connection between the identification results of PBG-SCM and that of original SCM.
>
> For the mentioned literatures, their results rely on stronger assumptions. For literature [1], it belongs to a branch (mainly for structure identification) which restricts the function class (linear non-Gaussian model). Such restriction is not acceptable in the field of disentangled representation learning, since the real functions are usually nonlinear. For literature [2], it introduces a strong structural assumption: guaranteed number of pure children, which is critical for successful structure identification but hard to satisfy for general SCMs. Remember that there is no free lunch.
>
> **For weakness 2:**
>
> We understand that you view the minimality assumptions as “not mild”, since it cannot be satisfied by an arbitrarily picked model. We respect your point of view, but please listen to ours. By proposing Proposition 5.1, we have shown that there always exists a valid model satisfying minimality assumptions, given invertibility and independence are already satisfied. We say “mild” since there are abundant models satisfying minimality assumption, and don’t need to worry about the existence of such models (find an invertible and independent one, and then apply Proposition 5.1).
>
> About the necessity of our assumptions, we list some examples for explanation. Consider a basis model $v_1=[s_1, z], v_2=[z, s_2]$. A case only violating invertibility:
>
> $\hat{s}_1=[s_1, c], \hat{z}=z, \hat{s}_2=s_2$, $c$ can be random noise;
>
> A case only violating independence:
>
> $\hat{s}_1=s_1+s_2, \hat{z}=z, \hat{s}_2=s_2$;
>
> A case only violating minimality:
>
> $\hat{s}_1=constant, \hat{z}=[z, s_1], \hat{s}_2=s_2$;
>
> All above cases can lead to the same observational distribution, thus the assumptions are necessary (we omit the structural equations for $v_1$ and $v_2$ here, since they are easy to get).
>
>
> **For weakness 3:**
>
> This is not a weakness, on the contrary, it is a contribution. Previous works usually omit the discussion of the importance of known dimension and simply assume it in their experiments. Exactly, we point out by our theory that, known dimension should not be a default setting, and need be highlighted both in the experiment setup and the theory. Please notice, we did not propose new identification algorithm as our contribution.
>
> **For weakness 4:**
>
> Considering the limited space, we have not included a proof sketch in the main text. We will consider adding one after arranging other contents. For the experiment, we have added a new experiment on controllable generation of images in the updated version (see Appendix A.9). Further experiments on real data may be unrealistic currently, since new identification algorithm is required to tackle the unknown dimension problem, which would be a challenging future work.
>
> **For question 1:**
>
> The definition of SCM has many equivalent versions, such as a tuple with 2 elements [1], 4 elements [2], or 6 elements [3]. We adopt an equivalent definition that emphasize the graph structure $\mathcal{G}$, while $\mathcal{F}$ are the functions in the assignments, and $\mathcal{P}$ is the joint probability. Note that $\mathcal{G}$ and $\mathcal{F}$ constitutes the original assignments.
>
> [1] Kaddour et al. 2022. Causal Machine Learning: A Survey and Open Problems
>
> [2] Moraffah et al. 2020. Causal Interpretability for Machine Learning - Problems, Methods and Evaluation
>
> [3] Bongers et al. 2021. Foundations of Structural Causal Models with Cycles and Latent Variables

---

### Official Review · Reviewer_aNXT · 2024-11-03

**Soundness:** 2
**Presentation:** 2
**Contribution:** 2
**Rating:** 3
**Confidence:** 4

**Summary:**

This work proposes an identifiability theory for nonlinear causal models with latent variables using only single-domain data. A minimality condition that has been implicitly used in most previous work has been formalized and leveraged for identification.

**Strengths:**

1. The considered problem is important and of practical significance.

2. The assumptions have been discussed in detail with examples.

**Weaknesses:**

1. The theory and methodology in this manuscript appear closely related to those in [1] for several reasons:

   - Both manuscripts decompose the original SCM into basis models, with very similar structures. For instance, Fig. 2 in [1] illustrates a basis model nearly identical to the one in this manuscript.
   - The identifiability theorems for the basis model share many similarities, differing primarily in the minimality assumption. While this manuscript makes the minimality condition explicit, [1] assumes known dimensions, implicitly ensuring minimality. Furthermore, the subspace span assumption in [1] is likely valid asymptotically, which may minimize substantial differences between the identifiability results in the two works.
   - For the general model, both manuscripts employ iterative identification of basis models to achieve global identifiability, following a similar approach.

Given the substantial overlap in the problem setting, theoretical results, and identification procedures, it may be helpful to further highlight the manuscript's unique contributions relative to [1], as the novel aspects are not yet sufficiently pronounced in the current version.


2. The notation could be clarified for better readability. For instance, $j\\&2^l$ in line 189 is difficult to interpret within the given context.

3. There are some questions for both the proofs and experiments. Please kindly refer to the questions for details.


[1] Kong et al., Identification of nonlinear latent hierarchical models, NeurIPS, 2023

**Questions:**

1. Could you please explain more on line 777, where it states that $\hat{v}_2$ does not change since the derivative of $v_2$ w.r.t. $s_1$ is zero?

2. The proof of Theorem 1 appears to establish only the nonexistence of certain edges, without addressing the existence of others. Including this additional aspect might provide a complete proof.

3. For evaluating general PBG-SCM identification, would it be more appropriate to use a metric that assesses component-wise identifiability, such as the mean correlation coefficient (MCC)? Since Theorem 2 establishes component-wise identifiability, relying on a metric like $R^2$, typically used for block-wise identifiability, may not fully capture the relevant evaluation criteria.

4. In all theorems, the generating functions are assumed to be invertible. However, in experiments, MLPs are used for the generation of synthetic data. Since MLPs are typically not invertible, it seems that the experiments cannot rigorously verify the theorems?

5. Could you please elaborate further on why previous methods require assumptions that there exists no connection among observed variables? I understand that this is necessary for the identification of latent structure, but not so sure about whether this is the case if we only aim to identify the latent variables. Perhaps elaborating it based on the concrete methods you mentioned in line 156 would be helpful.

---

> ### Author Response · Authors · 2024-11-27
>
> **For weakness 1:**
>
> Our work is significantly different with [1]. As discussed in line 109-110, there is a main drawback in theory of [1], i.e., their subspace-span assumption is about a relation between *two* models, making it infeasible to guide the design the identification algorithm (since the ground truth model is unknown). In contrast, our assumptions are all about the properties of *one* model, and we suggest to design restrictions for the learned model accordingly. The basis model may look similar, since it is a basic building block for larger models. However, there are differences between our basis model and [1]: we assume independence among all latent variables, while [1] allows dependence between $z$ and $s_2$; we aim to identify all latent variables, while [1] only aims to identify $z$. For the general model, our work is completely different from [1]: while [1] focuses on hierarchical structure (grow mainly in height), we retain two layers and only expand the number of variables (grow only in width). Meanwhile, the approaches are also completely different, the iterative nature of the proof is just a common math technique, there is nothing similar in the details.
>
> **For weakness 2:**
>
> The expression $j$&$2^l\neq 0$ means, in the binary representation of $j$, the $l$-th digit is $1$ (from right to left). We have added a new figure in the updated version to help understanding, see Appendix A.1.
>
> **For question 1:**
>
> There is no directed path from $s_1$ to $v_2$ in the model, so that $s_1$ does not influence $v_2$.
>
> **For question 2:**
>
> There are totally 9 edges in Fig. 7 (right), and we prove that 6 of them do not exist. So that the only 3 edges left are $s_1\to \hat{s}_1$, $z\to \hat{z}$, $s_2\to \hat{s}_2$, which are what we expect for identification.
>
> **For question 3:**
>
> We don’t really understand this question. We are focusing on block-wise identification (multi-dimension) rather than element-wise (single-dimension), what do you mean by “component-wise”? We follow [1] to use $R^2$, which in our view is reasonable. For “mean correlation coefficient”, do you mean the averaged Pearson correlation coefficient (PCC)? PCC is not suitable here, since it can only measure linear correlation, while we allow for nonlinear correlation here.
>
> **For question 4:**
>
> We have explained this in line 362-364. We ensure the parameter matrix is of full rank, and use invertible function Tanh as activation, so that the invertibility of MLP is guaranteed.
>
> **For question 5:**
>
> Here is an example for explanation. Consider a model with observed variables $v_1$ and $v_2$, if there exists an edge from $v_1$ to $v_2$, i.e. $v_2=f_2(v_1, \cdots)$, then replace $v_1$ with its structural equation: $v_2=f_2(f_1(PA(v_1)), \cdots)$. Run this process recursively, till all inputs of this equation are latent variables. Now you get a new SCM with no edge from $v_1$ to $v_2$, while keeping the observational distribution unchanged. In conclusion, there is always an equivalent model without directed paths among observed variables. Allowing such paths thus makes the model unidentifiable, unless including more structural assumptions, which is out of the scope of this paper.

---

### Official Review · Reviewer_EYVz · 2024-11-03

**Soundness:** 2
**Presentation:** 3
**Contribution:** 2
**Rating:** 3
**Confidence:** 3

**Summary:**

Identifiability analysis of latent (causal) variables is a fundamental challenge in causal/disentangled representation learning. This work introduces a straightforward powerset bipartite graph, which defines an equivalence class of general causal models capable of generating the same observed distribution. We present identifiability analysis tailored to this specific graph structure and extend the analysis to provide deeper identifiability insights. Simulations are conducted to validate the theoretical claims.

**Strengths:**

Identifiability analysis of latent variable models is both challenging and fundamental.

The identifiability analysis for this specific graph structure seems to be robust.

The relationship between the unknown nature of latent dimensions and the minimality condition is interesting.

**Weaknesses:**

As a theory paper, I take the details of claims very seriously. However, I found some claims to be overly casual, lacking reasonable justification.

1) The claim that 'multi-domain data are usually hard to acquire' seems overly strong. I do not think this is the case. Additionally, the references cited to support this point (e.g., Matsuura & Harada, 2020; Creager et al., 2021) primarily propose methods for handling multi-domain problems rather than the difficulty of acquiring such data, and therefore do not support the claim.

2) The claim that 'any underlying structural causal model (SCM) can be reduced to an equivalent SCM with a powerset bipartite graph (PBG) structure' also seems overly strong. I understand that, in latent space, generating the same observed data can result from multiple graph structures, leading to a large equivalence class. This is a key reason why non-identifiability results often arise, and many works aim to address this challenge. Given this, it seems reasonable that some transformations between potential graph structures could result in a standardized structure. However, the claim that 'any underlying SCM can be reduced to a powerset bipartite graph' appears unsupported. At the very least, the authors should provide rigorous theoretical backing for this claim.

3) Additionally, as an initial step in this work, it would be helpful to clarify which graph structures can be reduced to a powerset bipartite graph, which cannot, and what the associated costs of this reduction might be. Furthermore, if the powerset bipartite graph structure is critical, it would be valuable to present real applications that justify its importance.

4) The identifiability results in Theorem 1 appear to be sound. However, some implicit assumptions should be explicitly stated. For example, in reviewing the proof for Theorem 1, I noticed that it presumes mutual independence across all dimensions of each latent variable (e.g., z[i]⊥z[j] for the vector z). However, this assumption—that each zi​ is mutually independent—is not clearly stated in Theorem 1 itself. It is crucial to clearly list all assumptions to avoid ambiguity.

5) In general, non-identifiability is common in nonlinear ICA. However, applying sparsity constraints to the mixing process from latent to observed data can achieve identifiability, as demonstrated in this work. Prior studies, such as Zheng, Yujia, Ignavier Ng, and Kun Zhang's 'On the Identifiability of Nonlinear ICA: Sparsity and Beyond' (NeurIPS 2022), provide in-depth analysis on leveraging sparsity for identifiability. A clear comparison between this work and prior approaches is essential but appears to be missing here, making it difficult to fully understand this work's unique contributions.

6) Again, the experiments on real data are not directly provided, which significantly weakens the claim regarding the importance of the powerset bipartite graph.

**Questions:**

See Weaknesses.

---

> ### Author Response · Authors · 2024-11-27
>
> **For weakness 1:**
>
> We don’t view this claim as overly strong. Acquiring multi-domain data is at least harder than single-domain data, this is exactly the reason why some works try to infer the domain label or propose solutions without the need of multi-domain data. The two references are supportive in our view, we list some original text from their corresponding papers:
>
> Matsuura & Harada, 2020: “However, most datasets, such as those collected via web crawling, are a mixture of multiple latent domains, and it is difficult to know the domain labels.”
>
> Creager et al., 2021: “such environment labels are unavailable at training time, either because they are difficult to obtain or due to privacy limitations.”
>
> **For weakness 2:**
>
> In our view, the explicit reduction process provides a constructive proof itself. We realize that it may not be convincing enough, so we provide a proof in the updated version of our paper, see Appendix A.1 for it.
>
> **For weakness 3:**
>
> Given an SCM and an observed variable set, the reduction is possible as long as there are no directed path among observed variables. The associated cost may be losing information of unobserved endogenous variables (since they are originally unknown and unlikely to be identified), as well as fine-grained structures (since similar exogenous variables are concatenated). To understand the real applications of PBG, we give a simple example here: given 2 observed variables (such as image and its caption), can we discover the common factors that influence both (semantic information), and factors that influence only one of them (painting style, writing style)? This leads to a requirement for identifying $2^n$ latent variables ($n=2$ in this case), distinguished by whether they influence each observed variable.
>
> **For weakness 4:**
>
> Independence across all dimensions of each latent variable is not required by our theorem. We admit that the use of word “presume” may cause confusions. In fact, we have proved in the first paragraph of the original proof of Thm. 1 that, for a model that does not satisfy dimensional independence, it can be converted into an equivalent one that satisfies, without influencing the identifiability result. Due to such confusion, we have revised this part in the updated version.
>
> **For weakness 5:**
>
> We did not include much discussion with works on element-wise ICA, since they are typically not applicable in our block-wise identifiability settings. We are willing but currently unable to make a detailed comparison with works discussing sparsity, since this requires expanding our theory to element-wise ICA, which in our view is a new work. As far as our current understanding is concerned, our PBG structure provides a different kind of sparsity, such that each latent variable can be separated by the “difference” operation over the parent sets of each observed variables, while previous sparsity condition uses the “intersection” operation. Since the “difference” operation is strictly stronger than the “intersection” operation ($A\cap B=A-(A-B)$), our PBG structure may allow for more identifiable cases.
>
> **For weakness 6:**
>
> We provide a new controllable generation experiment on images in our updated paper, you can find it in Appendix A.9.

---

### Official Review · Reviewer_xSdi · 2024-11-04

**Soundness:** 2
**Presentation:** 2
**Contribution:** 2
**Rating:** 5
**Confidence:** 3

**Summary:**

The paper's main contribution is the introduction of a novel identifiability theory for latent variables in nonlinear causal models, using only single-domain data. It proposes a reduction process converting general causal models into powerset bipartite graph structures, proving that latent variables can be identified under mild conditions: invertibility, independence, and a new minimality condition. This theory provides a framework for designing algorithms to uniquely identify latent variables in scenarios with limited data. One main issue is that it claims that every model can be converted to a powers bipartite graph, but did not give any proof. Another weakness is that in all the examples in the paper, observable variables are children of latent variables, which is quite restricted and unrealistic.

**Strengths:**

The problem that the authors try to attack, i.e. latent causal discovery is crucial in causal discovery. It helps uncover hidden factors influencing observed data, enabling more accurate modeling of causal relationships. Identifying latent variables can improve understanding of complex systems, enhance predictive accuracy, and support tasks like disentangled representation learning, domain adaptation, and generalization to distributional shifts. Without addressing latent variables, causal models may miss key drivers, leading to incomplete or biased conclusions.

**Weaknesses:**

There are two main issues of the proposed approach.

(1) The conversion of SCM to a power-set bipartite graph is not proved to be correct.

(2) The assumption that observable variables are children of latent variables are too restrictive and inpractical.

**Questions:**

See above.

---

> ### Author Response · Authors · 2024-11-27
>
> **For weakness 1:**
>
> We did not provide a proof since the result is relatively obvious in our view. As there is confusion about it, we have added a proof in Appendix A.1 in the updated version.
>
> **For weakness 2:**
>
> Viewing observable variables as children of latent variables is typical in the field of latent variable identifiability [1-2], since the goal is to discover the underlying factors that influence observable variables. The only exceptions may be those allowing directed paths among observable variables, which we regard as future work and do not weaken the contribution of this work. We also provide a new controllable image generation experiment in the updated version (Appendix A.9), see if it helps for your understanding.
>
> [1] Kong et al. (2024) Identification of Nonlinear Latent Hierarchical Models.
>
> [2] Brady et al. (2023) Provably Learning Object-Centric Representations.

---

### Author Response · Authors · 2024-11-27
**Notification for updated paper**

We thank all reviewers for your efforts. We have updated our paper (modified content is in blue color), and suggest future discussion on the new version. Noticeable modifications include:

1.	We add an experiment on controllable generation of images in Appendix A.9.
2.	We add a proof regarding the validity of the SCM reduction process in Appendix A.1.
3.	We add a figure explaining the numbering rule in our SCM reduction and general model in Appendix A.1.
4.	We update part of the proof of Thm. 1 for better clarity in Appendix A.3.

---

### Meta-Review · Area_Chair_Mu9T · 2024-12-19

**Metareview:**

This paper tackles the important problem of latent variable identifiability in nonlinear causal models using single-domain data, proposing a framework based on powerset bipartite graph structures. While all reviewers think the topic is significant, there are still some concerns, including insufficient rigor in claims, restrictive assumptions limiting generality, overlap with prior work, and reliance on synthetic data with limited real-world validation. The authors’ revisions address some concerns but leave others, particularly around novelty and practical impact, unresolved. While the paper presents promising ideas, it requires more rigorous refinement.

**Additional Comments On Reviewer Discussion:**

There has been limited discussion for this submission. Reviewers have raised concerns about insufficient rigor in the claims, restrictive assumptions that limit generality, overlap with prior work, and reliance on synthetic data with minimal real-world validation. With scores of 5, 3, 3, and 3, the consensus is clear.

---

### Decision · Program_Chairs · 2025-01-22

Reject